# A New View on Risk of Typhoon Occurrence in the Western North Pacific

Kelvin. S. Ng[1], Gregor. C. Leckebusch[1]

[1]School of Geography, Earth and Environmental Sciences, University of Birmingham, Birmingham, UK

*Correspondence to:* Kelvin S. Ng (k.s.ng@bham.ac.uk)

**Abstract.** To study high impact tropical cyclone (TC) is of crucial importance due to its extraordinary destruction potential that leads to major losses in many coastal areas in the Western North Pacific (WNP). Nevertheless, because of the rarity of high-impact TCs, it is difficult to construct a robust hazard assessment based on the historical best track records. This paper aims to address this issue by introducing a computationally simple and efficient approach to build a physically consistent, high impact TC event set with non-realised TC events in the THORPEX Interactive Grand Global Ensemble (TIGGE) archive. This event set contains more than 10,000 years of TC events. The temporal and spatial characteristics of the new event set are consistent to the historical TC climatology in the WNP. It is shown that this TC event set contains ~100 and ~77 times more Very Severe Typhoons and Violent Typhoons than the historical records, respectively. Furthermore, this approach can be used to improve the return period estimation of TC-associated extreme wind. Consequently, a robust extreme TC hazard assessment, reflective of the current long-term climate variability phase, can be achieved using this approach.

## 1 Introduction

Increasing frequency and intensity of extreme meteorological events in the recent decades (IPCC, 2012) and increasing number of human population and assets located in risk-prone regions (Desai et al., 2015) lead to an increase of risk to humans and economic loss potentials from natural hazards e.g., tropical cyclones, with potentially disastrous consequences. For example, in the period between 1$^{st}$ January and 18$^{th}$ October 2018, total typhoon-related direct economic losses in China is evaluated to exceed 67 billion RMB (roughly 8.3 billion Euros) (Chinese Meteorological Administration, CMA, 2018). While natural hazards impact on all society stakeholders, governments are crucial in disaster risk reduction (DRR) because of their ability to implement necessary DRR-related policy and ability to allocate resources to appropriate parties (Shi, 2012). Governments have various options for DRR investments, for example, post-disaster relief and risk financing. Using cost-benefit analysis for a case study of typhoon disasters in China, Ye et al. (2016) showed insurance premium subsidies has the highest benefit-cost ratio. This is because premium subsidies increases penetration rate of an insurance program, i.e. more protection is offered by the private sector and the risk is transferred to the private sector (Glauber, 2004). Thus, development and application of effective financial instruments for risk transfer is important.

Other than classical (re-)insurance solutions, parametric insurance solutions have been developed for test cases in areas of corn yield (Sun et al., 2014) and livestock (Ye et al., 2017) for Southeast Asia and China in recent

years. Swiss Reinsurance Company Ltd. (Swiss Re) insured several municipal governments in Guangdong Province, China, through parametric insurance solution (Lemcke, 2017). Parametric insurance requires no physical damage assessment after an event. As soon as a certain threshold (i.e. trigger point) is exceeded, the insured party receives the agreed compensation from the insurer. Thus it has low administrative cost and quick disbursement. However, it is a challenge to determine a robust trigger point. It is because it would require a reliable typhoon hazard assessment for the region of interest. A current common approach is to generate a large typhoon event set (e.g. equivalent to 7,000 years of real world data) based on historical track data using stochastic approach (e.g. Vickery et al., 2000; Emanuel, 2006; Emanuel et al., 2006; Rumpf et al., 2007, 2009; Lee et al., 2018; Jing and Lin, 2020). There are two potential downsides with the stochastic approach: (i) such typhoon event set would be biased toward the past events, and the frequency-intensity distribution of the event set might not be the same as the underlying frequency-intensity distribution, and (ii) the storms in the typhoon event set which are created by stochastic approach are not necessarily physically consistent. As just surface footprints are stochastically modelled from existing tracks, there is no check whether those stochastically modelled events are physically possible and how they could be realised in a fully dynamical consistent view, i.e. fulfilling all known physical relations and derived constraints by the means of physical laws. Consequently, the amount of unrealistic physical properties due to the oversimplified stochastic simulation is unknown and laws of physical interactions are potentially ignored. Consequently, the trigger point derived from the common approach may not be optimal. This means insurees could be either over- or under-compensated by the insurer.

A method to increase number of extreme weather events is to make use of ensemble prediction system (EPS). Osinski et al. (2016) used European Centre for Medium-Range Weather Forecasts (ECMWF) EPS to build an event set of European windstorms. Osinski et al. (2016) pointed out there are two types of storm events produced by EPS: (i) modified EPS storm (MEPS), and (ii) pure EPS storm (PEPS). MEPSs are storms with modifications in the EPS which have real-world counterpart. PEPSs are storms in the EPS which have no real-world counterpart, i.e. unrealised. PEPSs are independent events and the number of PEPSs increases as the lead time increase until the model has no memory of the initial conditions. Thus one can form an event set of extreme weather event by using TC related PEPSs. Osinski et al. (2016) demonstrated that reliable statistics of storms under the observed climate conditions can be produced based on EPS forecasts.

Building upon the results of Osinski et al. (2016), a new approach to construct a large data volume, physically consistent TC event set is presented in this study. This event set is constructed by applying an impact-oriented windstorm tracking algorithm (WiTRACK; e.g. Leckebusch et al., 2008) to a multi-model global operational ensemble forecast data archive, the THORPEX Interactive Grand Global Ensemble (TIGGE) (Bougeault et al., 2010; Swinbank et al., 2015). The data volume of TIGGE is about 40,000 to 50,000 years. The event set consists of all non-realised TC events which were forecasted by EPS of different centres, this event set is referred to as the TIGGE PEPS (TPEPS) event set. In this study, we show the TPEPS event set has much higher information content: more TC events and more extremely high impact TC events than historical or reanalysis-based TC event set. The TPEPS event set can be used to produce a robust TC hazard assessment and to determine a robust trigger point for parametric typhoon insurance.

In this paper, we first present a computationally simple, inexpensive and efficient method to construct a physically consistent, high information content TC event set using only the 6-hourly surface wind speed field of

EPS forecast model outputs. Then we analyse the characteristics of the TPEPS event set. Validation of the new
method is done by comparing with the event set which is constructed using reanalysis data. The added values of
this new approach are also discussed and presented. The paper is organised as follows: data sets which are used
in this study are described in Section 2. Section 3 outlined the method that has been used to construct the TPEPS
event set. Results and discussions including validation and investigate the characteristic of the TPEPS event set
are presented in Section 4. A summary and conclusions can be found in Section 5.

**2 Data**
6-hourly instantaneous 10-m wind speed data in different data archives mentioned below are used in this study
because it is highly related to TC wind damages. The domain of this study covers the Western North Pacific
(WNP), east and south-east Asia spanning from 90° E to 180° E and 0° N to 70° N. The Japanese 55-year
Reanalysis (JRA-55) (Kobayashi et al., 2015) from 1979 until 2017 (resolution of 1.25°×1.25°) is used for
validation of the TPEPS event set. JRA-55 (1979-2014) is also used in parameter selection in TC identification
algorithm, construction of Logistic Regression Classifier (LRC) (Sect. 3.2.2), and the data in 2015-2017 are used
for validation of LRC. ERA-Interim (ERA-I) (Dee et al., 2011) is also used in the construction of LRC.
The TIGGE data archive (Bougeault et al., 2010; Swinbank et al., 2015) is used in the construction of
the PEPS TC event set. The TIGGE data archive has been used extensively in the study of TC activity forecast
(e.g. Vitart et al., 2012; Belanger et al., 2012; Halperin et al., 2013; Majumdar and Torn, 2014; Leonardo and
Colle, 2017; Luitel et al., 2018). TIGGE data archive consists of ~8-15-day ensemble forecast data from 10
numerical weather prediction centres with about 11-50 members each. In this study, only perturbed forecast
outputs of EPS from selected centres are used and they are CMA, ECMWF, Japanese Meteorological Agency
(JMA), and National Centers for Environmental Prediction (NCEP) (cf. Table 1). These four data sets are chosen
because they are the state-of-the-art NWP models, which is used by four leading synoptic weather forecast centres,
and they are the most complete dataset in the archive for the study period 2008-2017. Model configurations and
model updates are documented online at https://confluence.ecmwf.int/display/TIGGE/Models. ECMWF EPS is
a variable resolution EPS, i.e. days 1-10 were run at a higher resolution than days 11-15. For computational
efficiency, ECMWF EPS outputs are regridded into a lower resolution grid of 0.5625° × 0.5625°. The resolution
of the selected data sets ranges from 0.5625°×0.5625° to 1.25°×1.25°. Forecast lead time of each forecast outputs
ranges from 216 to 384 hours. Only forecast outputs, which are initialised during the main typhoon season, i.e.
15 May-30 November, are considered. The resultant TPEPS TC event set has data equivalent to more than 10,000
years of TC model data of the current climate state.
Many studies have evaluated the performance of these EPSs in forecasting TC activities in various ocean
basins. In general, EPSs underestimate TC intensity especially for coarse resolution models (Hamill et al.,
2010; Magnusson et al., 2014). TC track and genesis forecast error exists in EPS and these errors increase as lead
time increases (Buckingham et al., 2010; Yamaguchi et al., 2015; Zhang et al., 2015; Xu et al., 2016). While
ECMWF EPS forecast would occasionally have abnormal TC track forecast errors (i.e. track forecast error that is
extremely large) and might struggled with developing a warm core in the short range forecast (Majumdar and
Torn, 2014; Xu et al., 2016), ECMWF EPS appears to have better performance in TC track forecast than other
EPSs (Yamaguchi et al., 2015; Zhang et al., 2015; Xu et al., 2016). Yet, a full assessment of the respective skill
in models is not in the scope of this study. For the dedicated purpose of this study, an examination for biases in
the underlying climatological features as provided by a time- and ensemble-aggregated view of the data set is
presented in Sect. 4.1.

116        The International Best Track Archive for Climate Stewardship (IBTrACS) v03r10 (Knapp et al., 2010)

is used for validation and identification of TC events in reanalysis and TIGGE data archive. It contains all of the
available best track records from different centres around the globe up to year 2017. Since only part of the best
track records of year 2017 are archived in this version of IBTrACS, best track data from Joint Typhoon Warning
Centre (JTWC) is used for year 2017.

**3 Methods**
**3.1 Identification and characterisation of TC-related windstorms**
For identification and characterisation of TC-related windstorms, an impact-oriented tracking algorithm is used –
WiTRACK (Leckebusch et al., 2008; Kruschke, 2015). Befort et al. (2020) adapted the algorithm to TCs and
showed WiTRACK is well capable to identify high impact TC events in WNP, using coarse resolution reanalysis
product, with comparable quality to more data intensive algorithms. A brief description of the general procedure
to track a windstorm using WiTRACK is as follows: (i) clusters with wind speed above the local threshold are
identified for each of the 6-hourly time step of the input dataset, (ii) clusters with size smaller than a predefined
threshold (*minarea*) are excluded, (iii) clusters identified in each 6-hourly time step are connected to a track using
a nearest-neighbour criterion with consideration of the size of the cluster, and (iv) events with lifetime less than 8
6-hourly time steps are removed. Majority of the settings of WiTRACK are identical to Befort et al. (2020),
including the use of local $98^{th}$ percentile wind speed as local wind threshold, except in this study *minarea* is chosen
to be 15,000 km$^2$. The $98^{th}$ percentile wind speed is chosen because over 90% of loss events with losses above
3,000 million RMB can be identified by WiTRACK as demonstrated by Befort et al. (2020). The value for
*minarea* is chosen based on a series of sensitivity studies for parameter selection. The output of WiTRACK
contains information about the characteristics of all identified windstorm events, including size of the windstorm
at any given 6-hourly time step, the overall footprint of extreme wind associated with the windstorm events, and
storm severity index (SSI; Leckebusch et al., 2008). These information are used in the identification of TC-related
pure EPS windstorm events (Sect. 3.2). As discussed in Sect. 2, TC intensity is generally underestimated by EPS
and model resolution is known to be a limiting factor (Bengtsson et al., 2007; Hamill et al., 2010; Magnusson et
al., 2014). One of the advantages of using WiTRACK is that it does not use raw wind speeds, instead, it uses $98^{th}$
percentile relative exceedance for tracking. This means that even if the simulation wind speed of TC is
systematically weaker than historical observations, the $98^{th}$ percentile climatological wind in the models should
also be lower than the observed $98^{th}$ percentile climatological wind. A TC will still be tracked by WiTRACK as
long as there exists a $98^{th}$ percentile exceedance wind cluster. Consequently, a bias due to resolution does not
have significant impact on WiTRACK as the tracking algorithm serves as a bias correction in this sense (detailed
discussion on the impact of weaker wind speed in model outputs on WiTRACK can be found in Osinski et al.
(2016)). Furthermore, it can be shown that, within the study area, the 98th percentile relative exceedance of the
4 models, which we used to construct the TPEPS TC event set, have similar behaviour (i.e. similar to Figure 2 of
Osinski et al. (2016)). Consequently, individual PEPS TC event set can be combined to form a large PEPS TC
event set, i.e. TPEPS TC event set.

### 153      3.2 Identifying TC-related pure EPS windstorm events

WiTRACK identifies windstorm events of all kind, including MEPS TCs, PEPS TCs, MEPS extratropical
cyclones and PEPS extratropical cyclones. Therefore additional requirements are needed to identify typhoon-
related PEPS TC events. Four post-processing procedures are used: (i) Geographic Filter (GF), (ii) Logistic
Regression Classifier (LRC), (iii) MEPS TC Identifier (MTI), and (iv) Detection at Initialisation Filter (DIF).

### 158      3.2.1 Geographic Filter (GF)

GF was first introduced by Befort et al. (2020). It aims to remove non-TC-related windstorms, e.g. extratropical
cyclones, cold surge outbreaks during the winter monsoon, and equatorial disturbances, from the event set by
excluding windstorm events which solely identified north of 26° N and east of 100° E, and latitudinal position
exclusively south of 10° N. Befort et al. (2020) found this filter can reduce the false alarm rate (i.e. the ratio
between number of identified non-TC related windstorms and total number of detected windstorms) of TC
identification in JRA-55.

### 165      3.2.2 Logistic Regression Classifier (LRC)

In order to reduce computational cost and increase computational efficiency, the classical methods to determine
whether the atmospheric disturbance is a TC or non-TC via cold/warm core determination (e.g. Hart,
2003; Strachan et al., 2013) are not used because these methods require multiple variable fields which increase
computational cost significantly. Instead, a statistical learning approach, logistic regression classifier (LRC), is
used to determine whether the windstorm event is related to a TC or not. Details and background information of
LRC can be found in Hastie et al. (2009) and the *caret* package in R is used for LRC training (Kuhn et al., 2018;
available online at https://github.com/topepo/caret/). LRC is trained using the track characteristics of the event in
the JRA-55 and ERA-Interim event set (1979-2014) as explanatory variables (Table 2). This combination of
training set is chosen based on preliminary studies of constructing an optimal classifier using different
combination of training set. In order to avoid issues that are associated with collinearity, a stepwise Variance
Inflation Factor (VIF) selection method is used to identify independent variables. Variables with VIF value larger
than 5 are excluded. 17 variables have been chosen to use in the construction of LRC (Table 3). Variables that
relate to changes in storm position, lifetime of a storm, and mean wind field structure appear to be the most
important variables in the LRC. This is expected as the typical trajectory, duration, and structure of TCs and other
windstorms are very different. Validation using JRA-55 event set (2015-2017), which has 49 TC events and 47
non-TC events, have shown that the accuracy of the LRC is about 90% with low rate of false positives and false
negatives.

### 183      3.2.3 MEPS TC Identifier (MTI)

Since there are many replicated events of forecasted historical TCs (i.e. MEPS) in the operational forecast archive,
it is necessary to remove these events from our event set to avoid biases toward historical events. Instead of using
the criteria suggested by Osinski et al. (2016), a set of strict criteria (MTI) is used in this study. This can ensure
the statistics and climatology of TPEPS event set is not biased toward historical events. The MTI eliminates
forecast of MPES TC events where the forecasts of those MPES TCs were initialised (i) before, and (ii) after the
time of MPES TC genesis (hereinafter type 1 and type 2 forecast events respectively). A similarity index (*SI*) (Eq.
1) is used to eliminate type 1 forecast events:
$$d_i = \begin{cases} d_\text{thres} - d & d < d_\text{thres} \\ 0 & d \geq d_\text{thres} \end{cases},$$  (1a)
$$SI = \frac{\sum_i^{t_\text{overlap}} d_i}{d_\text{thres} \times t_\text{overlap}},$$  (1b)
where $d$ is the great circle distance between position of historical TC and position of TIGGE TC at the overlap
time step $i$, $d_\text{thres}$ is the maximum tolerance of $d$, $t_\text{overlap}$ is the number of overlap time steps in which both historical
TC and TIGGE TC existed and it must be larger than 4. Events with *SI* larger than *SI*$_\text{thres}$ are considered as MPES
TC events. A series of sensitivity study have been done for determining the optimal choice of parameters (not
shown) and the most optimal setting is $d_\text{thres}$=900 km and *SI*$_\text{thres}$ = 0.1. Type 2 forecast events are found if the
separation distance between the position of historical TC and the TIGGE TC at any point of their overlap time is
less than 400 km. This threshold is determined by the minimum separation between historical TCs and TC in
JRA-55 event set.
**3.2.4 Detection at Initialisation Filter (DIF)**
Any events that are detected at the time of model initialisation are removed following Osinski et al. (2016). It is
because these events are likely to be related to pre-existing disturbances or structures that leads to their
development. The removal of these events ensures the TPEPS event set is independent of any pre-existing weather
patterns.
**3.3 Adjustment procedure**
More than one windstorm event could be found within a close proximity of each other over the WNP. Since the
clustering algorithm in WiTRACK does not have a maximum size restriction on the cluster, multiple windstorm
events in close proximity could be identified as one windstorm event by WiTRACK. An additional procedure is
used to separate these merged windstorm events. This is an iterative procedure which would check whether all
of the grid boxes at each 6-hourly time step of the windstorm are within 1,000 km radius from the centre of the
windstorm cluster. If any of the event grid boxes are outside the 1,000 km radius, it will first remove these grid
boxes and recalculate the centre of event cluster. This procedure is repeated until there is no change in the centre
of cluster. This procedure addresses windstorm event with unrealistically large impact area and event SSI (ESSI).
The event SSI (ESSI) is defined as
$$\text{ESSI} = \sum_t^T \sum_k^K \left[ \left( \max\left(0, \frac{v_{k,t}}{v_{98,k}} - 1\right) \right)^3 \times A_k \right]$$  (2)
where $v_{k,t}$ is the wind speed at grid box $k$ and time step $t$, $v_{98,k}$ is the climatological 98th percentile wind speed at
grid box $k$, $A_k$ is the area-dependent weight. Summation is done over all time steps and all grid boxes affected by
the windstorm. The threshold radius is chosen to be 1,000 km because typical size of TC wind field is smaller
than a circle of 1,000 km radius (Lee et al., 2010; Chan and Chan, 2011).
**4 Results and discussions**
**4.1 Statistics and Validations**
In this section, we present validation of our TPEPS TC event set by comparing the climatological features as
provided by a time- and ensemble-aggregated view of the TPEPS TC event set to the historical/reanalysis based
event set. A historical TC is said to be detected in a forecast model if there exists a TC counterpart in the forecast
model, which is similar to the historical TC as identified by the MTI (c.f. Sect. 3.2.3). The detection rates of
historical TCs which are detected in different forecast outputs, i.e. CMA, ECMWF, JMA, and NCEP, are 91.2%,
94.7%, 89.4%, and 90.7%, respectively, whereas only 54.2% of historical TCs in the period of 2008-2017 are
detected in JRA-55 (Table 4). Since WiTRACK is a wind threshold exceedance based detection scheme and the
98[th] percentile wind speed value of JRA-55 within the tropical WNP is similar to these selected TIGGE data (Fig.
1), this implies JRA-55 underestimates the wind speed of wind field of TCs, which is in agreement with Murakami
(2014). This also shows these selected TIGGE outputs provide a better representation of the atmosphere. Total
515,712 TC related windstorm events are detected in the selected TIGGE data set. ~38.5% of the all TPEPS
events are PEPS TC events (Table 5). Percentage of total TC windstorms as PEPS TCs can be treated as a proxy
to quantify the forecast skill of the model. For example, NCEP has 47.1% of TC windstorms as PEPS TCs
whereas JMA has 26.5%. This indicates the NCEP model generates more "wrong" forecast than JMA however
these "wrong" forecasts are physically possible. Yet, examining the forecast skill of models is not the focus of
this study and the rest of the discussion focuses on the TPEPS TC event set.
Figures 2 and 3 show the spatial pattern and temporal variability of the number of TC which are first
detected for each day, respectively, of the TPEPS and JRA-55 event sets. While individual model might have
bias in certain spatial and temporal domain, for example the region with the highest track density of JMA is at the
eastern WNP in Fig. 2d in comparison to other models, and NCEP failed to capture the peak activity prior 2012
in Fig. 3, the overall patterns of the TPEPS event set match the JRA-55 event set. This is expected because (i)
TC formation depends on the environmental conditions and initial disturbance (Gray, 1977; Ritchie and Holland,
1997; Nolan, 2007). During the period of active TC season, environmental conditions over WNP are usually
favourable for TC formation but often there is no suitable disturbance in the region. Since EPS simulates the
chaotic behaviour of the atmosphere, it would forecast disturbances which would be possible to form but not
realised in the real atmosphere. Hence PEPS TCs can be formed in those period of time over WNP. And (ii) the
trajectory of TCs depends mainly on the large scale environmental flow of the region (Chan, 2010). This implies
PEPS TCs would also follow the typical trajectory of real TCs given that the large scale flow is correctly
represented in the forecast models. Thus, in general the spatial and temporal patterns of the TPEPS event set
match the patterns of JRA-55 event set. There are several possible reasons which lead to the differences in spatial
pattern between TPEPS event set and JRA-55 event set. The eastward bias in the track density appears to be a
common feature in many GCMs (e.g. Camargo et al., 2005; Bell et al., 2013; Roberts et al., 2020), this has also
been observed in seasonal forecast output (Camp et al., 2015). Finite simulation time has also contributed to this
bias as TC that forms in the region east of 150 °E would not have sufficient time to move into the western part of
WNP before the end of simulation time. Differences in number of tracks could also contribute to the differences
in spatial pattern as more diverse tracks would be captured in larger event set.
Some TPEPS events appear in locations where no historical TC event is observed (Figs. 2c and 2f).
While there is no historical TC event in some locations, this does not imply TC cannot occur in those regions.
The historical data, which cover 39 years of observations, may not have enough samples to construct a distribution
that can correctly represent the basic population (i.e. all possible TCs in the given climate). For example, the
occurrence of Tropical Storm Vamei that formed close to the equator (~1.4° N) does not satisfy the classical
"necessary but insufficient" conditions of TC formation, which are identified by Gray (1977) based on historical
observations. This shows TC can appear in historically "TC-free" region. Furthermore, from the statistical
perspective, the JRA-55 event set can be viewed as a subset which is randomly selected from the TPEPS event
set. To provide more evidence to support this view, we have conducted bootstrap resampling on the TPEPS event
set to obtain 10,000 sets of subsamples. Each set of subsamples has 668 events to mimic the number of events in
the JRA-55 event set. Uncentred pattern correlation between the track density of the JRA-55 event set and the
track density of each set of subsamples are calculated. In order to focus on the relevant entries, if the values of
track density of a grid box for a resampling set and the JRA-55 event set are both less than 1, such grid box is
neglected in the pattern correlation calculation. The mean, standard deviation, minimum and maximum of the
uncentred pattern correlation of the 10,000 set of subsamples are 0.9380, 0.0107, 0.8961, and 0.9697, respectively.
This suggests the spatial pattern of the JRA-55 event set is highly similar to some small random subsets of the
TPEPS event set. Thus, the JRA-55 event set can been seen as a subset which is randomly selected from the
TPEPS event set. On the other hand, it is not be possible to deduce the basic population (e.g. the TPEPS event
set) from a small sample set (e.g. the JRA-55 event set). Although the spatial distribution of the small set sample
is similar to the subsamples of the basic population and thus usable as one possible realisation of the basic
population, the small sample set does not contain all of the information of the underlying population. Furthermore,
the statistical estimate of extremes would also be different for the small sample set and the basic population.
Some of the examples of TPEPS TC tracks and impact footprints are shown in Fig. 4. The trajectory of
these TPEPS TC tracks is indistinguishable to historical TC trajectories in WNP. This shows these TPEPS TC
events are realistic and physically possible events. Figure 5 shows the climatological daily number distributions
of TC first detection for TPEPS TC event set and JRA-55 event set. Although the peak activities period of JMA
is slightly lagged behind and the over- and under-estimation of the peak of activity for CMA and NCEP are
observed, respectively, the seasonal cycle of TPEPS TC event set is well captured and this matches to the seasonal
cycle of the JRA-55 event set. This shows our new approach is capable to produce spatially and temporally
realistic events.
In general, the temporal evolutions of the number of first storm detections of TPEPS event set during the
integration time has an increasing trend in the short lead time followed by a roughly constant behaviour (Fig. 6).
In short lead time (i.e. close to initialisation of forecast), the true state of the atmosphere is well simulated by
forecast models, thus EPSs are likely to produce storms that actually occurred (i.e. MEPS storms) and less likely
to produce PEPS storms (Osinski et al., 2016). As lead time increases, more PEPS storms are produced due to
increasing uncertainty of the state and the chaotic behaviour of the atmosphere in EPSs. When EPS has no
memory of the initialisation state of the atmosphere, the probability distribution of formation of PEPS TCs
becomes a uniform distribution.
The overall impact of any storm is related to the many factors for example lifetime of the storm, the size
of the storm, and the intensity (or strength) of the storm (e.g. Vickery et al., 2000; Mori and Takemi, 2016; Kim
and Lee, 2019). Here we investigate whether there are systematic biases in the TPEPS TC event set which would
affect these quantities. The lifetime distribution of TPEPS TCs matches to the JRA-55 event set but proportionally
overestimates for short-lived TCs and underestimates for long-lived TCs (Fig. 7a). These differences are the
consequence of the finite simulation time in forecast models. If the same restriction (i.e. finite simulation time
window) is applied to the JRA-55 TC event set (grey shaded areas in Fig. 7), the lifetime distribution of TPEPS
TCs would be in good agreement to the JRA-55 TCs. Similar conclusion can be reached in the comparison of the
distribution of time required to reach lifetime maximum intensity (LMI) (Fig. 7b). However, finite simulation
time of EPSs cannot explain the difference in the distribution of impact area, which is the total area that has
experienced TC-associated extreme wind (i.e. larger than local climatological 98$^{th}$ percentile wind speed), between
TPEPS and JRA-55 event sets despite they have the same type of distribution (Fig. 7c). The difference in the
distributions of impact area maybe due to the fact that wind speed of the TC wind fields is underestimated in JRA-
55 as discussed above. Consequently, many weaker TCs, which would have small impact areas, are not detected
and thus they are not necessarily included in the JRA-55 TC event set.
**4.2 Robust TC hazard assessment**
To demonstrate the benefit of our approach, TC records in IBTrACS, JRA-55 TC event set, and TPEPS TC event
set are stratified into intensity classes according to their lifetime maximum intensity (c.f. Table 6). Since
WiTRACK is an impact-oriented, wind speed percentile based tracking scheme which tracks TCs with potential
impact (Befort et al., 2020), many of the low impact TCs (i.e. TCs in the Tropical Depression and Tropical Storm
(TD&TS) category) are not detected and thus not included in the TPEPS TC event set. Focusing onto the
categories of high impact TC, i.e. Typhoon (TY), Very Strong Typhoon (VST), and Violent Typhoon (VTY), the
TPEPS event set contains 302.14, 102.54, and 77.02 times more TY, VST, and VTY than the IBTrACS records,
respectively. This means our new approach can capture much more extremely high impact events such that a
more robust analysis of extreme TC events can be done.
The key advantage of this new approach is that it constructs a physically consistent and high information
content TC event set with good and realistic representation of the current climate state using a computationally
inexpensive algorithm. Since more physically consistent and physically possible TCs are included, more extreme
events can be captured in the TPEPS event set. Consequently, a robust TC hazard assessment can be obtained.
Some of the examples are presented in this subsection.
Figure 8 shows the location of first detection of TCs with LMI at least typhoon strength, which made
landfall within the given domain (105-180° E, 0-30° N) for TPEPS and JRA-55 TC event set. The spatial pattern
of the TPEPS TC event set (Fig. 8f) matches the spatial pattern of the JRA-55 TC event set. The data in the JRA-
55 TC event set are sparse and it does not provide sufficient information about whether TCs, which made landfall
in this region, are typically first identified in the WNP or in the South China Sea (SCS). The TPEPS TC event
set, on the other hand, provides a clearer picture and suggests events, which made landfall in this domain, are
typically first identified in the SCS and western WNP. This is consistent with the known climatology. As TCs
within the SCS and western WNP usually follow the western and northwestern trajectory and subsequently made
landfall over the Vietnam, south and southeast mainland China, Taiwan, and the Philippines.
Figure 9 shows the number of TC landfall events, which made landfall with at least typhoon strength,
with the focus of southern and southeast mainland China, and Taiwan. Much more landfall events have been
captured by TPEPS TC event set (11449) than the JRA-55 TC event set (100). The spatial distribution of TPEPS
TCs is in good agreement with the JRA-55 TCs with uncentred pattern correlation of 0.8345. TCs, which made
landfall with at least typhoon strength, are more likely to made landfall along the coast of the southern Fujian
Province and the eastern Guangdong Province than any other coastal area of South and Southeast mainland China.
Furthermore, higher TC landfall frequency is observed on the side of islands (i.e. Hainan Island and Taiwan)
which faces the open ocean than the other side of islands. This is consistent with observations. The TPEPS TC
event set also provides information about the frequency of TC landfall at locations where no landfall events had
observed in the JRA-55 TC event set, e.g. locations along the coastline of Guangdong Province. Furthermore, the
distribution of landfall intensity for TCs, which made landfall with at least typhoon strength, for the TPEPS TC
event set is very similar to the JRA-55 TC event set (the null hypothesis, i.e. the distributions are the same, is not
rejected at the 0.05 significance level of the two-sample Kolmogorov-Smirnov test).
**4.3 Application**
The TPEPS TC event set is constructed based on physical models, i.e. GCMs, which provide a good representation
of the atmosphere of the real world. The wind field associates to a TPEPS TC event is realistic and local effects,
such as local topography, have been taken into account. This implies the wind information of the TPEPS TC
event set can be used for estimates return periods of local extreme wind events associated with typhoon with high
confidence. Figure 10 shows the number of TC-related 6-hourly extreme wind (i.e. wind speed higher than the
local 98$^{th}$ percentile climatological wind speed) data entries in each of the grid box within Guangdong Province
in the Southern China. The JRA-55 TC event set can only construct a TC-related 6-hourly extreme distribution
with ~25 (inland) and ~325 (coastal) data entries whereas such distribution can be constructed with at least 500
to over 28,000 data entries using the TPEPS TC event set. This implies the estimated return period using the
TPEPS TC event set would be more reliable than using the JRA-55 TC event set and similarly the observation
data alone. This is of importance from the DRR perspective as wind speed values are used in practice to decide
on payments out of parametric insurance products (Swiss Re, 2016). Consequently, reliable wind-based trigger
points of typhoon parametric insurance can be determined. This will further improve the suitability and flexibility
of parametric insurance for DRR applications. Ultimately, this will improve the speed of post-disaster recovery.
A demonstration for such application is given below.
Four surface observation stations are chosen for this demonstration, they are Baiyun International Airport
(BAIYUN INTL; 23.392° N, 113.299° E; from 1945-2019), Baoan International Airport (BAOAN INTL; 22.639°
N, 113.811° E; from 1957-2019), Shanwei (22.783° N, 115.367° E; from 1956-2019), and Shangchuan Dao
(21.733° N, 112.767° E; from 1959-2019). For each selected surface station, the grid box of each EPS that
corresponds to the surface station is identified (Fig.11). Resolution of models is known to be a factor to limit the
wind speed of TCs (Bengtsson et al., 2007). This means for the same TC, the associated wind speed would be
lower in low resolution model and higher for high resolution model. In order to utilise the extreme wind

information from EPSs with different resolution, the cube of $98^{th}$ percentile relative exceedance of wind speed (EXCE) is used. Since EXCE is a ratio, it is a resolution independent quantity and the tail behaviours of the EXCE distribution for these models are similar, which is in agreement with Osinski et al. (2016). Information from different models can be combined using EXCE. EXCE entries, which correspond to TC in the TPEPS TC event set, are extracted for those grid boxes. This forms a set of "observations" of the impacts of high impact TCs at those grid boxes in the model space. We assume all of the EXCE entries are independent and identically distributed (iid) random variables. This is a reasonable assumption, due to the fast moving nature of TCs, diverse possible direction of the movement of wind field, and rapid decay of wind field over land for a 6-hour interval, local observations often have only one extreme wind observation for a TC event. In order to translate this information to the physical world, quantile mapping is used for mapping EXCE to the observed surface wind speed which exceeded local climatological $98^{th}$ percentile. Historical in situ surface wind data are obtained from the Integrated Surface Database (ISD) (Smith et al., 2011). Quantile mapping is done using the R package *qmap* (Gudmundsson et al., 2012; Gudmundsson, 2016). Due to different geographic configuration and climatology of each in situ observation station, different quantile mapping strategies have been employed. The optimal strategy is chosen based on minimisation of the root-mean-square-error (RMSE) of the quantile mapping output (see Gudmundsson (2016) for more details). Using above information, the return period-return level plot (using threshold exceedance approach) is constructed using the R package *extRemes* (Gilleland and Katz, 2016). For detail discussion of calculation of return period and return level, readers are referred to Elsner et al. (2006), Jagger and Elsner (2006), and Gilleland and Katz (2016). Figure 12 shows the return period-return level plot of four selected stations which are derived using our proposed approach with the TPEPS TC event set and using in situ observational data. The width of the 95% confidence interval which is calculated using our proposed approach is much sharper than the 95% confidence interval which is calculated using in situ observational data. In other words, the uncertainty can be reduced by using the TPEPS TC event set because more observations are used in the calculation.

The above application of the TPEPS TC event set can provide crucial information for the DRR community. As discussed in the introduction, typhoon parametric insurance can be an effective financial instrument for typhoon risk transfer. However, an effective typhoon parametric insurance requires a robust trigger point, which is determined by the meteorological information, e.g. wind speed. If the trigger point is too high, disbursements would not be made even if a catastrophic meteorological disaster has occurred, i.e. under-compensation; If the trigger point is too low, disbursements would be made even if no catastrophic event has occurred. Using the TPEPS TC event set, the estimated return period has smaller uncertainty than the estimation made by in situ observational data, such that an optimal trigger point for typhoon parametric insurance can be determined.

**5 Summary and Conclusions**

In this study, a new and efficient method, which addresses the critical issue in typhoon risk assessments – a robust methodology to determine the real frequency of TC occurrence with high socioeconomic impact potential by constructing a physically consistent TC event set, is presented. This is achieved by applying an objective impact-oriented windstorm identification algorithm – WiTRACK, on 6-hourly 10-m horizontal wind field of selected ensemble data set from a multi-centre grand ensemble data archive – TIGGE. While WiTRACK identifies major

events based on one meteorological variable only, it is capable of identifying events of general loss relevance as demonstrated by Befort et al. (2020). This implies the event set generated by our approach is in principle suitable for general TC risk assessments, as well as for an assessment of the hazards frequency-intensity distribution specifically. Several sensitivity tests with different parameter settings are done using JRA-55 data to obtain the optimal setup for WiTRACK. Since WiTRACK can identify all types of windstorm events, four post-processing procedures are used to identify PEPS TCs, these procedures include a geographic filter and logistics regression classifier. The TPEPS event set has the climatological spatial and temporal pattern of TCs which match to the historical climatological pattern of TC in WNP. More than 302, 102, and 77 times of TY, VSTY, and VTY, respectively, are found in the TPEPS TC event set in comparison to the IBTrACS record. A robust representation of extreme TC events in WNP can be obtained using the TPEPS TC event set because of the high number of physically consistent extreme events. Consequently, a robust hazard assessment of land-affecting TCs in the WNP can be produced using the event set constructed by this new method. Furthermore, the return-period of typhoon-related extreme wind events e.g. Typhoon Haiyan (2013) and Typhoon Mangkhut (2018), can be determined with sharper confidence intervals in a similar manner as Walz and Leckebusch (2019). As a result, policymakers and related stakeholders can improve the current typhoon related disaster reduction and mitigation strategy. Furthermore, a robust trigger point for parametric typhoon hazard insurance can be determined using our proposed approach by reducing the uncertainty of estimated return period of a meteorological extreme event. This will improve the suitability and flexibility of parametric insurance for DRR applications. Consequently, this will improve the speed of post-disaster recovery.

The TC event set constructed using the method described in this study has several unique properties in comparison to the TC event set constructed by other methods (Vickery et al., 2000; Emanuel et al., 2006; Rumpf et al., 2009; Kim and Lee, 2019):

(i) Many methods in the literature (e.g. Emanuel et al., 2006; Rumpf et al., 2009) use historical best track data to construct a spatial probability function that determine the genesis location of synthetic TCs and a parametric track model, that matches to the historical observations, to determine the movement of synthetic TCs. Consequently, these synthetic tracks are highly likely to be identified in the region where TCs were identified from the historical observations and highly rare in the region where TCs were never identified but physically possible. In contrast, TPEPS TCs are detected at any physically possible locations over the WNP. This means, besides the events, which are similar to the historical observations, the TPEPS TC event set also includes events that occur in the region where no historical event was observed. Consequently, the TPEPS TC event set provides an important and unique advantage for typhoon hazard assessment. In comparison to other methods to generate large TC event sets, our specific approach is limited mainly by the source of data used. The current TC event set constructed using medium range forecasts archived in TIGGE, is strictly spoken representative only for the current climate state. Any longer-term climate variability (e.g. multi-decadal fluctuations like the Pacific Decadal Oscillation (PDO)) and their impacts on any TC frequency-intensity distribution are not accounted for in this setting. Nevertheless, the presented approach would be equally applicable to data sets representing that kind of variability on longer time scales (e.g. decadal predictions or transient climate model simulations).

(ii) In the literature, the structure of wind field of synthetic TCs follows a predefined, analytical model, e.g. parametric vortex structure developed by Holland (1980) or modified Rankine vortex. For the TPEPS TC event

set, complex physical processes in GCMs determine the structure of wind field of TCs, therefore the structure of wind field of TCs is realistic. This is an advantage for robust wind hazard assessment of land-affecting TCs because the resultant wind field includes the complex atmosphere-land interaction which depends on the local topography. Consequently, the TPEPS TC event set can be used as addition observations for the estimation of return period of TC-related extreme wind as demonstrated above.

(iii) Many of the TC risk assessments are done based on wind risk, and/or wind-induced coastal risk but not TC-related precipitation risk (Vickery et al., 2000; Emanuel et al., 2006; Rumpf et al., 2009; Mendelsohn et al., 2012; Mori and Takemi, 2016; Marsooli et al., 2019; Kim and Lee, 2019). A reason is that historical damages due to TC-related wind are much better documented than TC-related precipitation damages (Emanuel et al., 2006). However, damages due to TC-related precipitation, e.g. flooding, should not be ignored. Based on the pay-out of the National Flood Insurance Program of the United States for the flood event of Hurricane Ike (2008), Smith and Katz (2013) estimated the insured flood damage as 5.4 billion USD. Furthermore, some of the high impact TCs in WNP have typical typhoon intensity but the amount of rainfall is extremely high, e.g. Typhoon Morakot (2009) (Wu, 2012). Since precipitation is one of the output variables of these medium range ensemble forecasts, precipitation-related impact can be examine by integrating the realistic precipitation information from forecast outputs into the TPEPS TC event set. Furthermore a spatial distribution of TC related hazard, e.g. extreme wind and extreme precipitation, of the TPEPS TC event set can be constructed using the notion of TC hazard footprint (Chen et al., 2018). Consequently, a more thorough typhoon risk assessment can be achieved. This is currently under our investigation.

In conclusion, the event set that we have constructed contains all necessary information for applications in the DRR context. This event set can improve the hazard component in an overall assessment of integrated TC risks (e.g. Sajjad and Chan, 2019) by providing a robust probability of occurrence of extreme TC event. Furthermore, using this event set, a robust trigger points of parametric insurance for the local hazard can be determined. Once such trigger points for the local hazard are available (including their uncertainty), the targeted application of parametric insurance products in disaster relief application is possible. Especially, when it comes to the evaluation of the basis risk. This study is merely the first step toward a statistically robust, full physical model based TC hazard assessment. The impact of TC-related extreme precipitation and storm surges can be integrated following the approach developed by Befort et al. (2015).

*Data availability.* JRA-55 (Kobayashi et al., 2015) and ERA-I (Dee et al., 2011) are freely available for academic use at the UCAR Research Data Archive: https://rda.ucar.edu/datasets. The TIGGE dataset (Bougeault et al., 2010; Swinbank et al., 2015) used in this study can be accessed through ECMWF server: https://apps.ecmwf.int/datasets/data/tigge/levtype=sfc/type=pf/. IBTrACS (Knapp et al., 2010) and ISD (Smith et al., 2011) are available at the United States National Centers for Environmental Information, National oceanic and Atmospheric Administration: https://www.ncdc.noaa.gov/ibtracs/index.php, and

https://www.ncdc.noaa.gov/isd, respectively. JTWC best track data used in this study is obtained from the United
States Navy Website: https://www.metoc.navy.mil/jtwc/jtwc.html?best-tracks.

*Author contribution*. KSN and GCL originated the idea, developed the methodology, performed data analysis, and
wrote the paper.

*Competing interests*. The authors declare that they have no conflict of interest.

*Acknowledgments.* The authors thank three reviewers for their helpful and constructive comments. The authors
thank Drs. D. Befort and M. Angus for valuable discussion. This work was supported by the Building Resilience
to Natural Disasters using Financial Instruments grant INPAIS (Integrated Threshold Development for Parametric
Insurance Solutions for Guangdong Province China, Grant Ref: NE/R014264/1, through Natural Environment
Research Council (NERC). The computations described in this paper were performed using the BlueBEAR HPC
service at the University of Birmingham.

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

**Tables**

| Centre | Number of members | Runs per day | Resolution | Implementation date | Forecast lead time (hr) |
|---|---|---|---|---|---|
| CMA | 14 | 2 (00, 12 UTC) | 0.5625°×0.5625° | 20070515 | 240 |
| | | 2 (00, 12 UTC) | | 20140805 | 360 |
| ECMWF | 50 | 2 (00, 12 UTC) | 0.5625°×0.5625° | 20061001 | 360 |
| JMA | 50 | 1 (12 UTC) | 1.25° × 1.25° | 20060301 | 216 |
| | 50 | 1 (12 UTC) | | 20130328 | 264 |
| | 26 | 2 (0, 12 UTC) | | 20140226 | 264 |
| NCEP | 20 | 4 (0, 6, 12, 18 UTC) | 1.0° × 1.0° | 20070327 | 384 |


**Table 1**. Information of selected data sources from TIGGE archive.

| Variables |
|---|
| Time average of area of cluster |
| Time average of longitude of cluster centre |
| Time average of latitude of cluster centre |
| Time average of maximum extent of cluster |
| Time average of mean wind speed |
| Time average of standard deviation of wind speed |
| Time average of minimum wind speed |
| Time average of maximum wind speed |
| Time average of longitude of location of maximum wind |
| Time average of latitude of location of maximum wind |
| Time average of storm severity index (SSI) |
| Standard deviation of time series of area of cluster |
| Standard deviation of time series of longitude of cluster centre |
| Standard deviation of time series of latitude of cluster centre |
| Standard deviation of time series of maximum extent of cluster |
| Standard deviation of time series of mean wind speed |
| Standard deviation of time series of standard deviation of wind speed |
| Standard deviation of time series of minimum wind speed |
| Standard deviation of time series of maximum wind speed |
| Standard deviation of time series of longitude of location of maximum wind |
| Standard deviation of time series of latitude of location of maximum wind |
| Standard deviation of time series of storm severity index |
| Number of 6-hourly time steps |
| Area of windstorm event footprint |
| Event SSI |
| Difference of latitude between the initial and final locations |
| Difference of longitude between the initial and final locations |
| Total distance travelled |


**Table 2**. List of explanatory variables which are initially considered in the LRC model.

| Variable | t-value |
|---|---|
| Difference of latitude between the initial and final locations | 12.5707 |
| Difference of longitude between the initial and final locations | 9.9983 |
| Time average of standard deviation of wind speed | 9.3709 |
| Time average of minimum wind speed | 8.5015 |
| Time average of maximum extent of cluster | 5.1416 |
| Number of 6-hourly time steps | 4.8719 |
| Standard deviation of times series of latitude of location of maximum wind | 3.4302 |
| Standard deviation of times series of mean wind speed | 2.3640 |
| Standard deviation of times series of area of cluster | 2.2447 |
| Event SSI | 1.9621 |
| Standard deviation of times series of maximum extent of cluster | 1.7922 |
| Time average of latitude of cluster centre | 1.4493 |
| Standard deviation of time series of SSI | 0.9980 |
| Standard deviation of times series of longitude of location of maximum wind | 0.9237 |
| Standard deviation of times series of standard deviation of wind speed | 0.7268 |
| Time average of longitude of location of maximum wind | 0.4204 |
| Standard deviation of time series of minimum wind speed | 0.2613 |


**Table 3**. List of explanatory variables and their associated t-value which are used in the construction of LRC.

| Year | IBTrACS | CMA | ECMWF | JMA | NCEP | JRA-55 |
|---|---|---|---|---|---|---|
| 2008 | 21 | 19 | 19 | 19 | 17 | 10 |
| 2009 | 22 | 20 | 20 | 20 | 14 | 10 |
| 2010 | 13 | 13 | 13 | 13 | 13 | 6 |
| 2011 | 21 | 19 | 20 | 17 | 19 | 14 |
| 2012 | 24 | 23 | 23 | 23 | 23 | 16 |
| 2013 | 29 | 28 | 28 | 27 | 28 | 15 |
| 2014 | 19 | 12 | 17 | 17 | 17 | 13 |
| 2015 | 22 | 20 | 21 | 20 | 21 | 17 |
| 2016 | 26 | 25 | 25 | 24 | 25 | 13 |
| 2017 | 30 | 28 | 29 | 23 | 29 | 9 |
| Total | 227 | 207 | 215 | 203 | 206 | 123 |
| Detection Rate | | 91.2% | 94.7% | 89.4% | 90.7% | 54.2% |


**Table 4**. (From the left) Annual number of historical TCs in IBTrACS (second column); Annual number of
historical TCs detected in the respective forecast models (third to sixth columns); Annual number of historical
TCs detected in JRA-55 (seventh column).

| Centres | Number of TC windstorms | Number of Pure EPS TCs | % of TC windstorms as pure EPS TCs |
|---------|-------------------------|------------------------|-------------------------------------|
| CMA | 39535 | 13322 | 33.7 |
| ECMWF | 215737 | 74091 | 34.3 |
| JMA | 56537 | 14964 | 26.5 |
| NCEP | 203903 | 96052 | 47.1 |


**Table 5**. Statistics of TCs in the selected TIGGE data.

| Intensity Class | IBTrACS | JRA-55 | TPEPS |
| --- | --- | --- | --- |
| TD&TS | 252 | 32 | 27643 |
| STS | 208 | 126 | 70759 |
| TY | 231 | 254 | 69794 |
| VSTY | 231 | 193 | 23686 |
| VTY | 85 | 63 | 6547 |
| Total | 1007 | 668 | 198429 |


**Table 6**. Number of TC records in IBTrACS, JRA-55 TC event set, and TPEPS TC event set, for different
intensity classes. The classes are Tropical Depression (TD) and Tropical Storm (TS), Severe Tropical Storm
(STS), Typhoon (TY), Very Strong Typhoon (VST), and Violent Typhoon (VTY). The intensity classes for
IBTrACS are defined according to WMO (2019). The intensity classes for JRA-55 TC and TPEPS TC are derived
from the WMO (2019) intensity classes by using quantile mapping of intensity records of JRA-55 TC and
IBTrACS records.
**Figures**

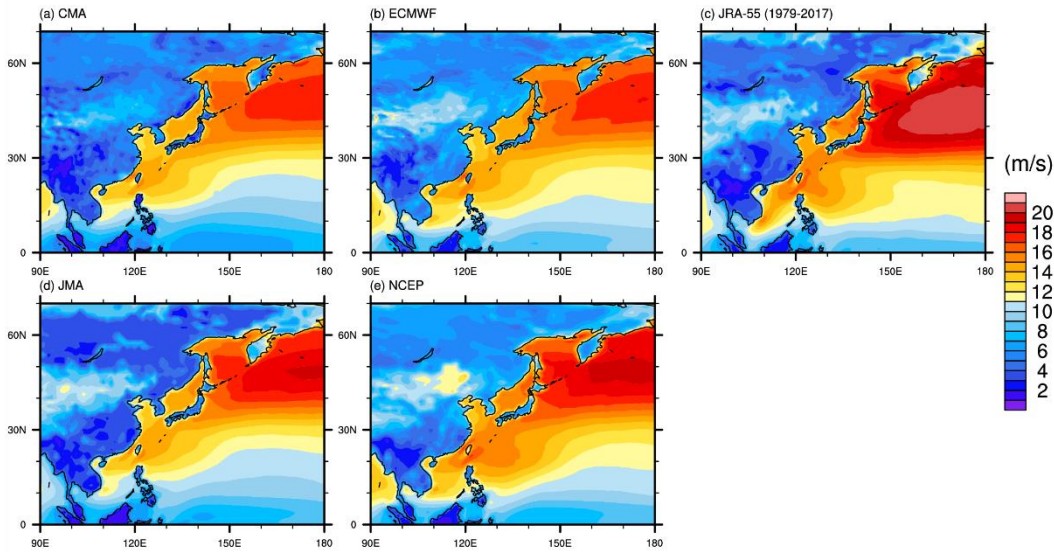


**Figure 1**. Local 98<sup>th</sup> percentile wind speed for each grid box in the region for TIGGE: (a) CMA, (b) ECMWF, (d)
JMA, (e) NCEP, and (c) JRA-55.

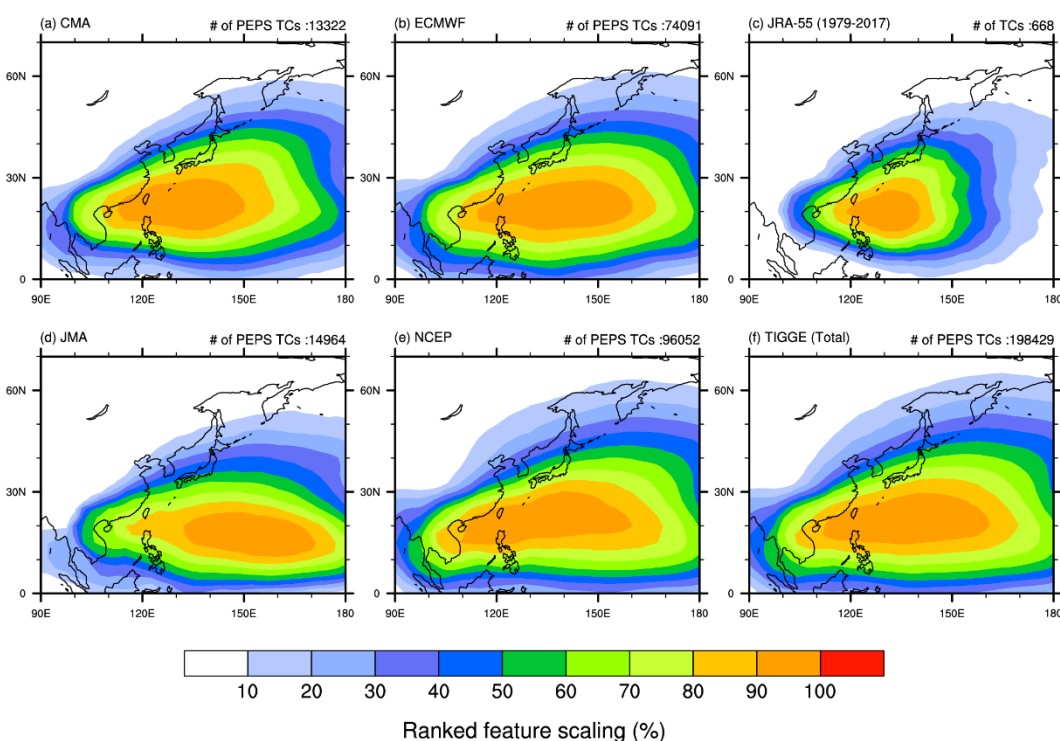


**Figure 2**. Ranked feature scaled track density (%) of different data sets: (a) CMA, (b) ECMWF, (c) JRA-55, (d) JMA, (e) NCEP, and (f) TIGGE total.  Number of TCs in the corresponding event set is stated on the top right of each panel.


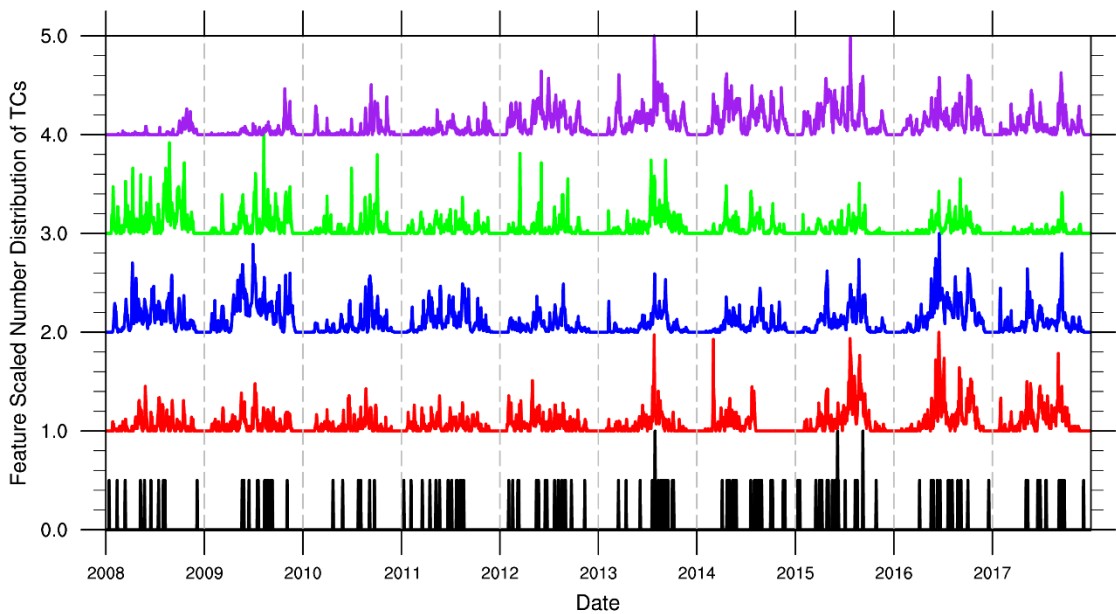


**Figure 3**. Feature scaled time series of number of TCs which are first identified in each day in the TPEPS TC event set (CMA: red, ECMWF: blue, JMA: green, NCEP: purple) and JRA-55 event set (black). For visual convenience, the time series of CMA, ECMWF, JMA, and NCPE are shifted by 1, 2, 3, 4, respectively.


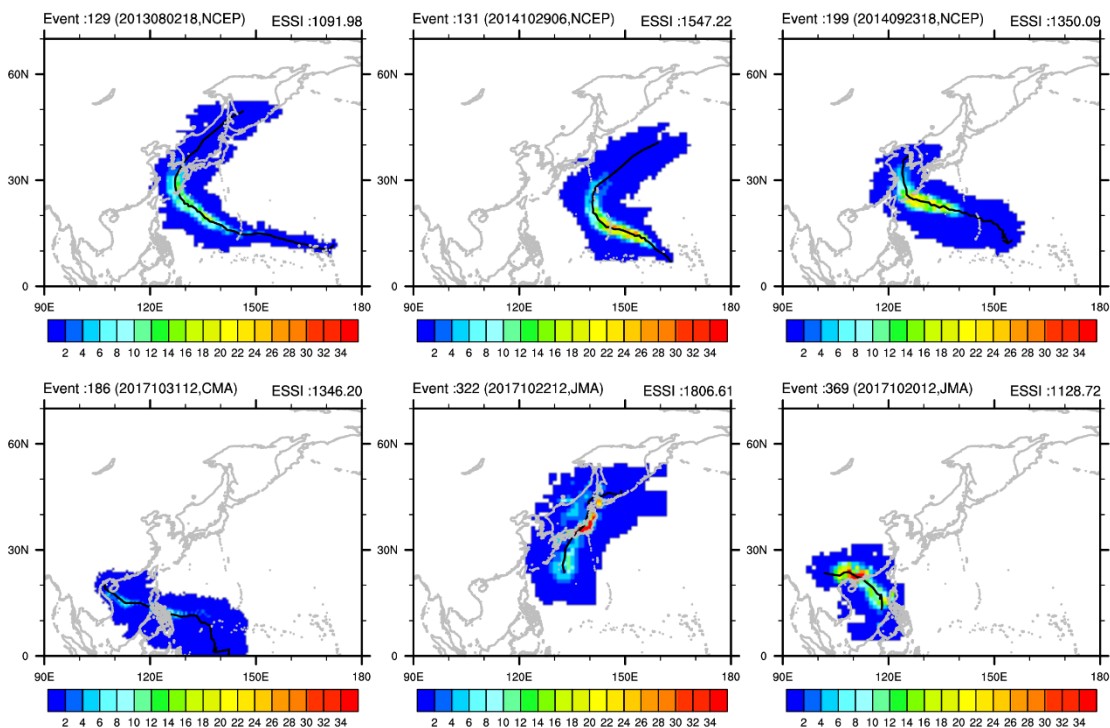


**Figure 4**. Some of the PEPS TC impact footprint (colour contours) and tracks (black line within the colour
contours) of the TPEPS TC event sets. The colour contours show the cumulative SSI of the PEPS TCs over their
respective lifetime at individual grid box. ESSI of each PEPS TC is shown on the top right of each panel.

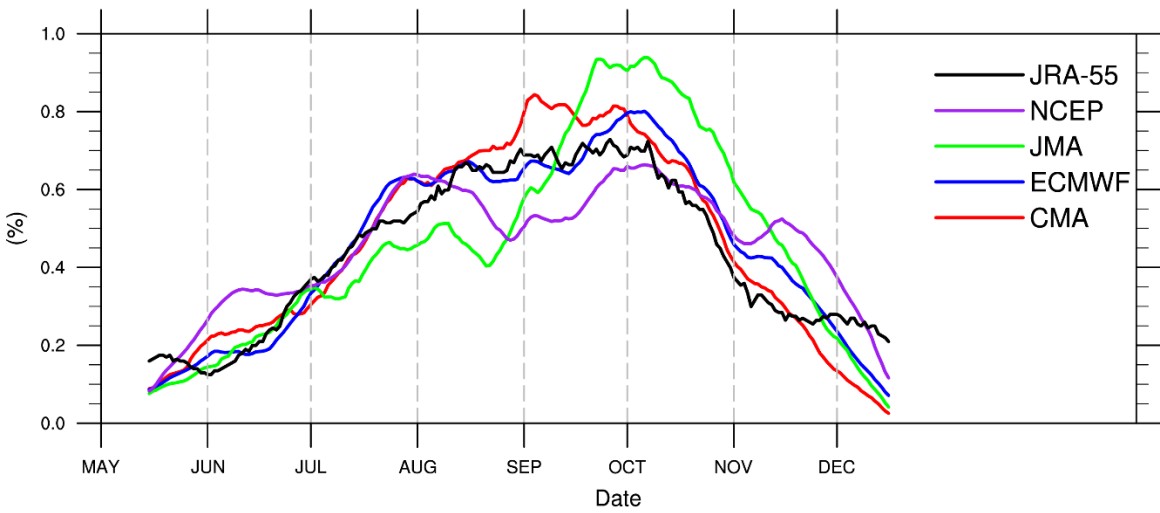


**Figure 5**. Climatological daily number distribution of TC first detection for TPEPS TC event set (CMA: red,
ECMWF: blue, JMA: green, NCEP: purple) and JRA-55 event set (black), i.e. the probability of TC being first
detected at a given day in the model. 30-day moving average is used in order to remove high frequency signal.

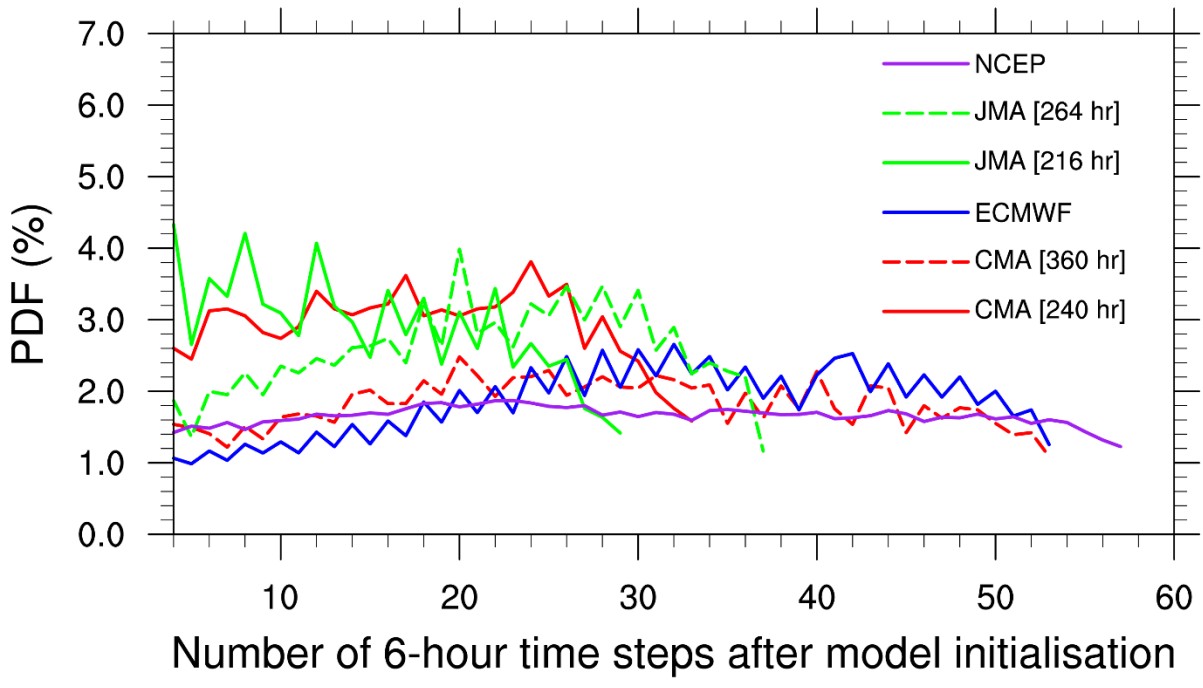


**Figure 6**. Temporal evolution of frequency of first storm detections of TPEPS event set (CMA: red, ECMWF: blue, JMA: green, NCEP: purple).

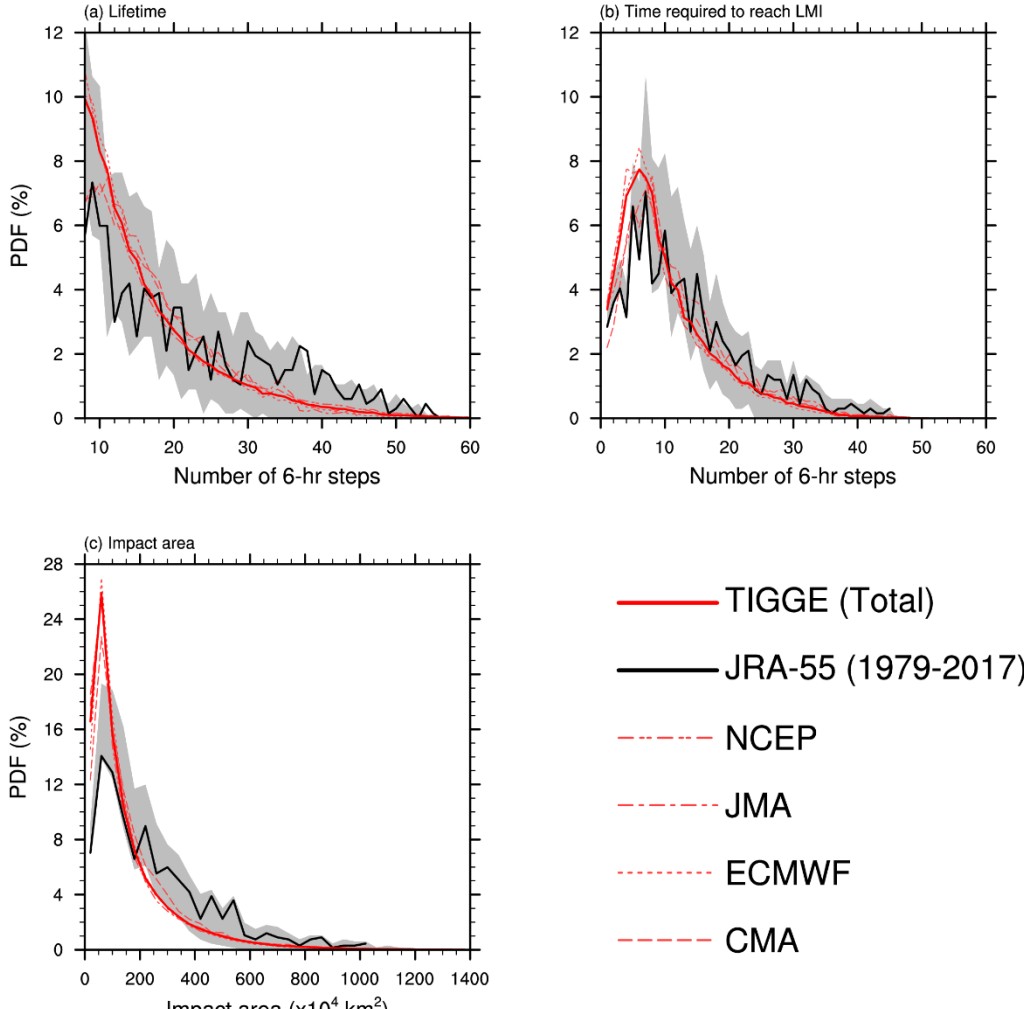


**Figure 7**. The distribution of (a) lifetime, (b) time required to reach LMI, and (c) impact area of TCs in TPEPS
TC event set (red lines) and JRA-55 event set (black line).  The grey area indicates the spread of the lifetime
distribution of JRA-55 if finite simulation windows are applied to the JRA-55 event set.


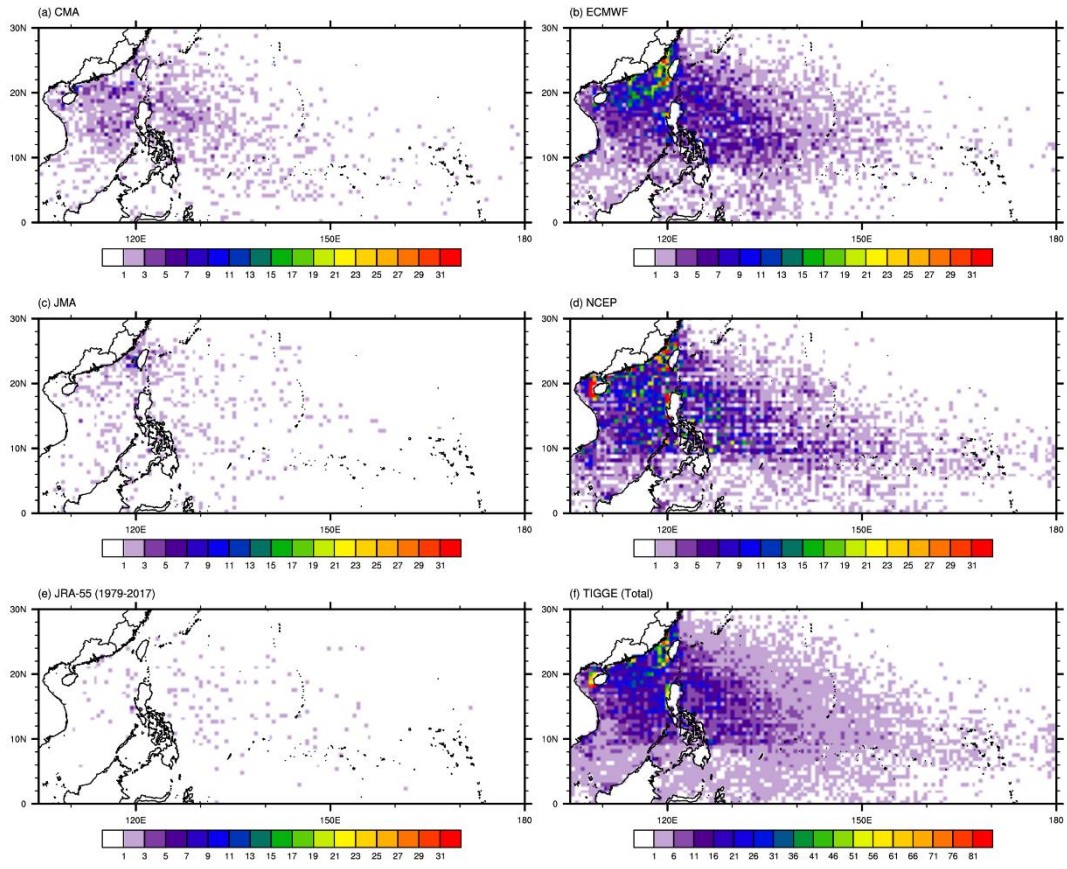


**Figure 8**. The spatial distribution of location of first detection of TCs (with LMI at least typhoon strength) which
made landfall within the domain 105-180 °E, 0-30 °N for TPEPS TC event set and JRA-55 event set.

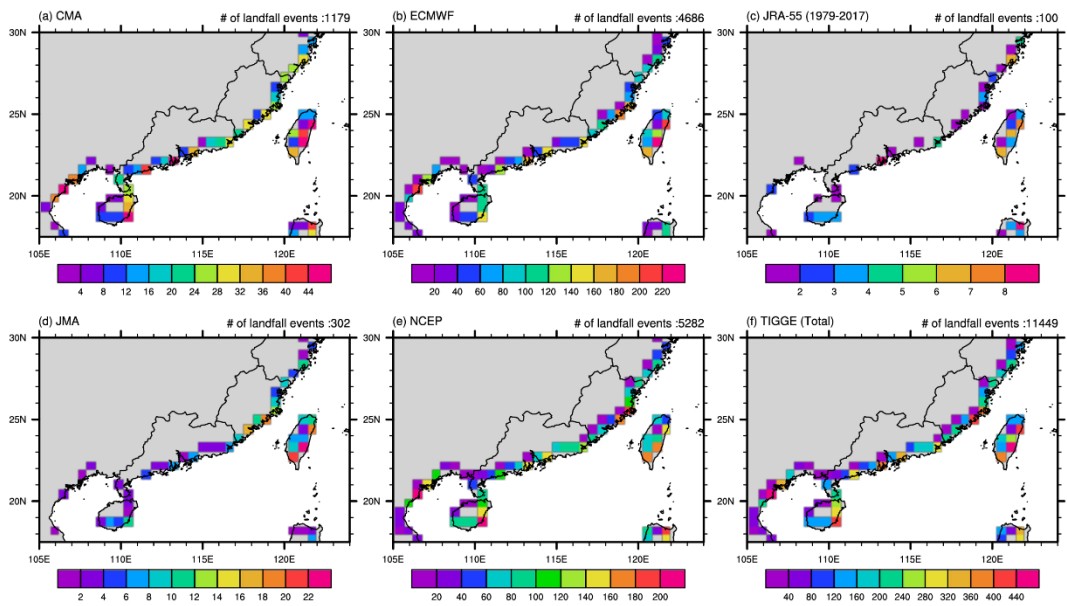


**Figure 9**. Spatial distribution of number of landfall events (landfall with at least typhoon strength) for TPEPS TC
event sets and JRA-55 event set (colours). The total number of landfall events in each panel is shown on the top
right of each panel.

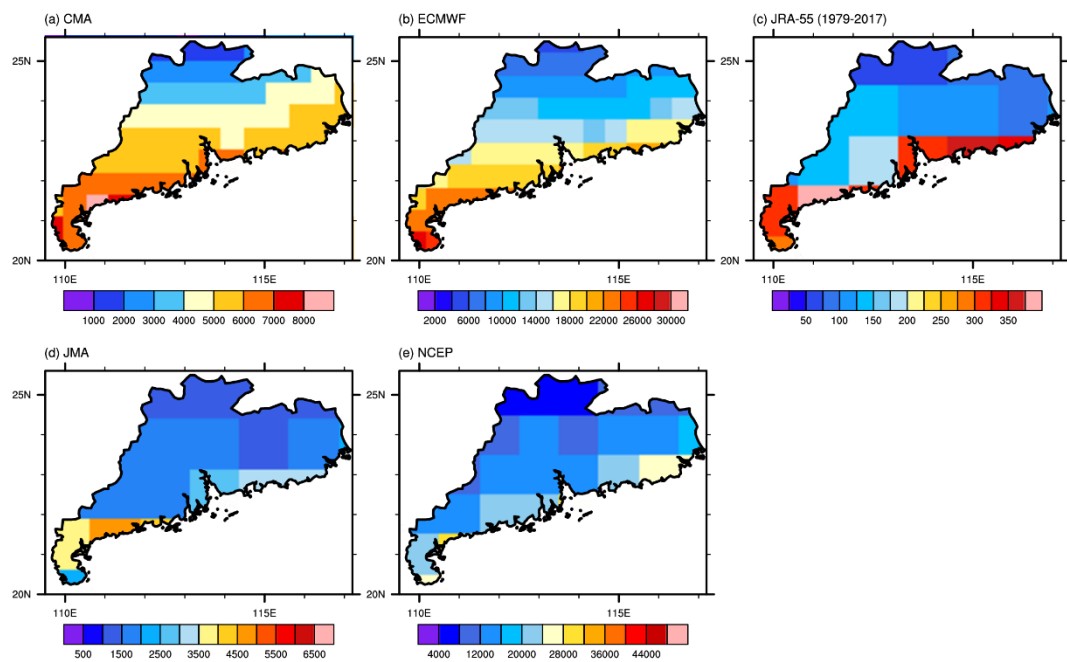


**Figure 10**. Number of TC-related 6-hourly data entries in each of the grid box in Guangdong Province, China,
for TPEPS TC event sets and JRA-55 event set.

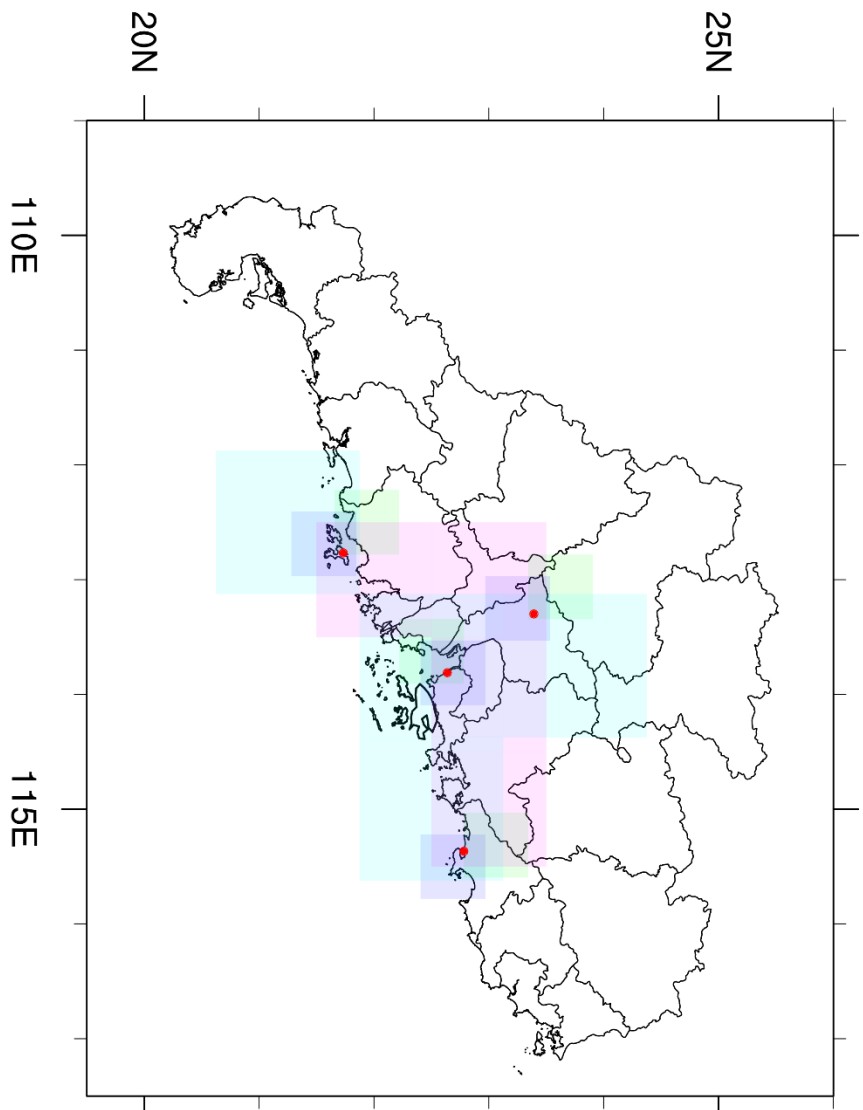


**Figure 11**. Locations of the selected surface observation stations (red dots) in Guangdong, China with corresponding grid boxes from 4 EPS outputs: CMA (green), ECMWF (blue), JMA (cyan), and NCEP (magenta). Information of prefectural boundaries is obtained from GADM version 3.6 Level 2 (available at https://gadm.org/data.html)


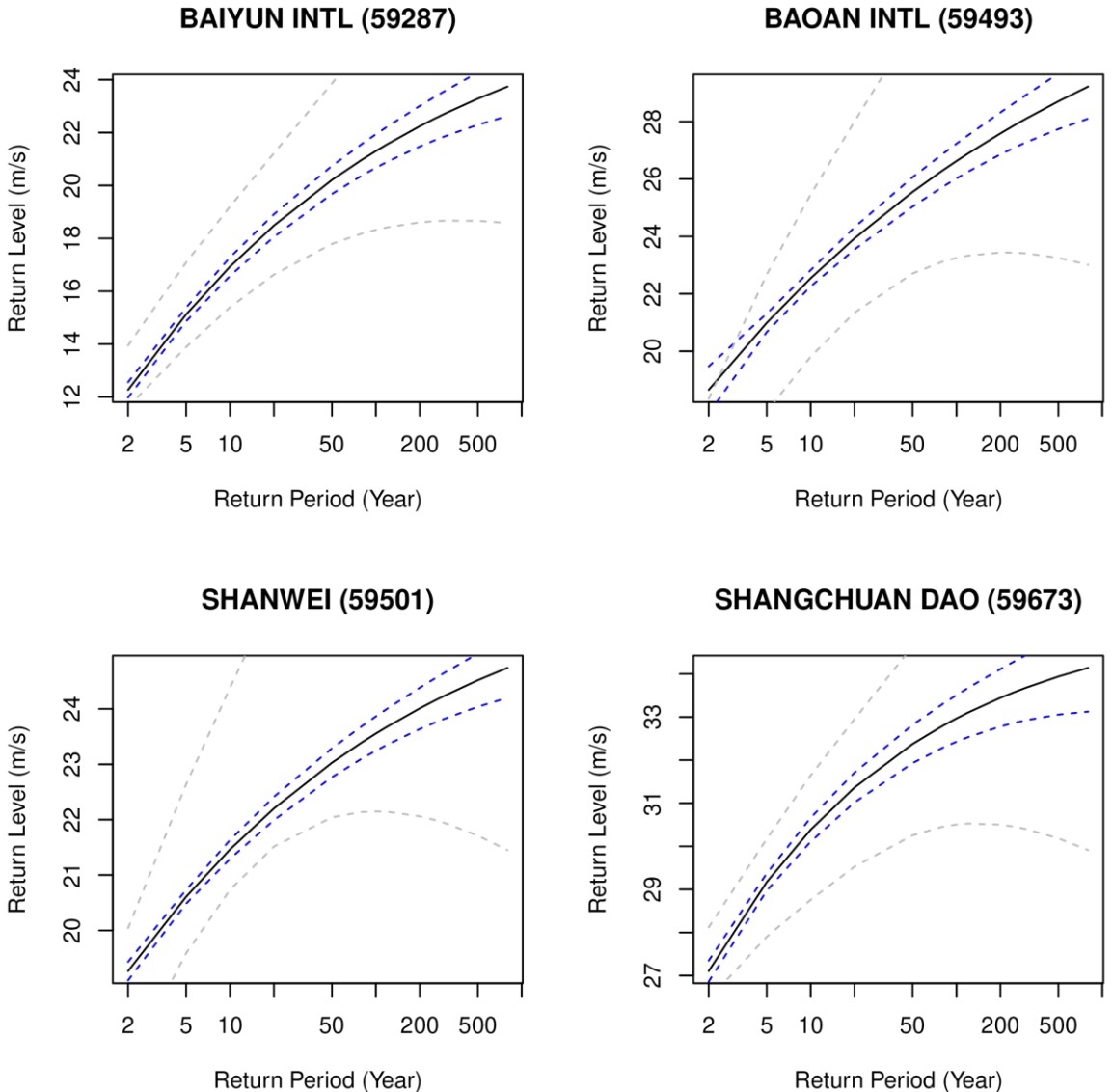


**Figure 12**. Return period-return level plot for 4 selected surface observation stations: Baiyun International Airport,
Baoan International Airport, Shanwei, and Shangchuan Dao. Black lines indicate the best estimate of return
period-return level. Blue lines indicate the 95% confidence interval calculated using TIGGE PEPS event set.
Grey lines indicate the 95% confidence interval calculated using in situ observations.