# Peer review of "A New View on Risk of Typhoon Occurrence in the Western North Pacific"

_Natural Hazards and Earth System Sciences, 2020_

## Referee Comment (RC1) · Anonymous Referee #1 · 3 Aug 2020

**Major issues**

You have explained that you only used 6-hourly surface wind speed (i.e., lines 67-69 and discussed in Section 4.2 as well), is your risk assessment model able to comprehend a full TC risk? What about TC-rainfall (you only mentioned about it in the last 2-3 sentences in Conclusions) and TC-induced storm surge, which are more significant than winds in many regions of the world. For example, the recent Hurricane Florence in USA and Mangkut in South China caused significant damaged due to surge and rainfall, respectively. Therefore, under such circumstances, what is the applicability of your proposed approach? You need to think about the generalization, replicability, and adoptability of your approach from a wider perspective and not only from the study area.

Recently, Sajjad and Chan 2019 and Sajjad et al. 2020 proposed typhoon risk frameworks based on TC hazard (wind-based similar to yours), vulnerability, and disaster resilience, which provides a comprehensive information on TC risk. They found that the Pearl River delta region in Guangdong (area primarily mentioned in your case, Line) is a statistically significant hotspot of TC risk. How do you see the usefulness of your method for such frameworks? A thorough discussion regarding this is necessary

Similarly, most of the discussion in the manuscript revolves around TC-hazard and neglects the vulnerability and resilience within the regions where TCs are making landfalls. For instance, you say on Lines 236-238 that overall impacts of a storm is related to many factors such as size, duration, and intensity. However, the impacts are not only related to TC-associated factors but vulnerability and resilience are also integral parts of overall impacts and risks associated with TCs, as discussed in Sajjad and Chan 2019 and Sajjad et al. 2020. How do you incorporate these characteristics within the TC risk discussion of yours?

- Sajjad, M., & Chan, J. C. (2019). Risk assessment for the sustainability of coastal communities: A preliminary study. Science of The Total Environment, 671, 339-350.
- Sajjad, M., Chan, J. C., & Kanwal, S. (2020). Integrating spatial statistics tools for coastal risk management: a case-study of typhoon risk in mainland China. Ocean & Coastal Management, 184, 105018.

Additionally, you need to detail the current limitations of your method. For example, how well this method could perform at higher resolution assessments, which are more important

for policy and decision-making in the context of DRR efforts? What are the future prospects of your study?

**Specific Comments**

**Lines 10-13:** The sentence is long and it is difficult to follow. Would be better to break it into two sentences, if possible.

Line 22: There is no such thing as "natural disaster" but only natural hazards. Disasters always involve human agency. Therefore, please avoid using this term and check the manuscript thoroughly for this issue. For further details, you are encouraged to see <a href="https://www.undp.org/content/undp/en/home/blog/2017/5/18/Natural-disasters-don-t-exist-but-natural-hazards-do.html">https://www.undp.org/content/undp/en/home/blog/2017/5/18/Natural-disasters-don-t-exist-but-natural-hazards-do.html</a>

**Line 34: you mean "livestock"?**

**Lines 145-149:** It is mentioned that VIF is used to resolve the issue of collinearity and 17 variables are selected to construct the final LRC model. How many total variables were included initially? Are the VIF values for all of these 17 variables less than the normal threshold (i.e., VIF  $\leq$  7.5)? It would be useful to add the VIF values of the final variables in Table 3.

**Lastly**, a thorough intermediate level editing is recommended to remove "several" grammatical and language errors throughout the manuscript.

---

## Referee Comment (RC2) · Anonymous Referee #2 · 21 Aug 2020

Manuscript number: NHESS_2020_74
Full Title: A New View on Risk of Typhoon Occurrence in the Western North Pacific

*General comments:*
This manuscript describes a new and efficient method to produce TC event set in the Pacific basin. The TC events are detected from an ensemble data archive TIGGE, using an objective impact-oriented windstorm identification algorithm WiTRACK. This dataset contributes to existing synthetic datasets (mostly statistical basin-wide methods) as in this dataset the TCs are detected in GCMs, so that the complex physical processes of TCs are captured and hence TCs are physically realistic. More extreme TCs are found in the dataset and these data will help overcome the shortage in observational record.  Overall, I think the results will be a nice contribution to the field of TC risk assessment. However, I have some basic questions about the methodology, the utility of certain results and I recommend major revisions prior to publication. Also, I would recommend careful editing of the manuscript. There are many terminologies in the manuscript and please make sure it is easy for readers to follow.

*Major comments:*
1) L191: The detection rates of historical TCs are reported here. However, will the detection algorithm produce more TCs? What fraction of TCs that the detection algorithm produce is real historical TCs?

2) The authors mention that one benefit of this dataset is "The TPEPS event set includes events which are unlikely but physically possible. This provides an important and unique advantage for typhoon risk assessment." Combined with Fig. 2, TC tracks in the detected dataset is very different with observations, and TPEPS tracks appear in locations with no historical tracks. If there is no historical track in some regions, are they supposed to be no storms or there can be storms but no storm has appeared in historical records due to the low probability? This needs to be explained.

3) The sensitivity and performance of four ensemble data archive are not well described. For example, in some dataset, the storms are much weaker than historical storms. And some models have biases in simulating extratropical cyclone transition. More explanations and descriptions of the data archive needs to be added. Also, how these biases would have an impact on the detection algorithm?

4) The authors have compared the TIGGE PEPS TCs with JRA-55 in terms of track density, landfall frequency, etc. How about other characteristics? For example, landfall intensity along coastline?

5)  Fig. 7 shows the difference between TIGGE PEPS event set and observation. In the text, you have mentioned possible reasons for these differences. Is there possible way to reduce these differences, for example in the detection algorithm, to also remove low-impact storms? Also, you mentioned the ESSI, is there a way to quantify this index?

*Minor comments:*

L41-42: more recent papers should be added. Such as the following two recent models:
- Lee, C.-Y., M. K. Tippett, A. H. Sobel, and S. J. Camargo, 2018: An environmentally forced tropical cyclone hazard model. Journal of Advances in Modeling Earth Systems, 10 (1), 223–241.
- Jing, R., and N. Lin, 2020: An environment-dependent probabilistic tropical cyclone model. Journal of Advances in Modeling Earth Systems, 12 (3), e2019MS001 975.

L45: I didn't understand the sentence 'the typhoon event set might not be physically consistent'. What is 'physically' consistent?

L79: "The domain of this study covers the Western North Pacific (WNP), east and south-east Asia spanning from 85 E to 195E and 15 S to 75 N." Why data around equator is also used? There is no TCs forming around equator.

L102-104: Is there a reason why an old version of IBTrACS is used?

L152: "the accuracy of the LRC is about 90%" What is the fraction of TC (or positive samples?) Does there exist issue of imbalanced data?

L197: "Percentage of total TC windstorms as PEPS TCs can be treated as a proxy to quantify the forecast skill of the model." In Table 5, NCEP is almost twice of that in JMA, what does this percentage mean?

L203: do you mean Fig 3? Also, more explanations should be added in the text. I can't understand this figure.

L203: In Fig. 2, all TPEPS are much more similar with each other, comparing with JRA-55. How to explain this?

Fig4: The tracks in black are very easily messed up with the map. Probably change the color of coastline.

Fig5: The y-axis is not clear to me, please add more explanation.

Fig8: The colored dots for single center are too light to see. If this figure is to show distribution, I would recommend not using same color bar for single model and for TIGGE total.

Fig9: It's hard to see the distributions are in good agreement, probably can change to annual frequency instead of total number of landfall events. Also, the correlation coefficients could be used to show the landfall frequency in all TIGGE dataset is positively correlated with JRA-55.

Fig12: I can see your points in showing the grey dashed lines. But the lower bound curves can not show the trend properly. I would recommend add 75% or 80% confidence interval to show that the trends are same, but TIGGE PEPS event set has much narrower bounds.

---

## Referee Comment (RC3) · Anonymous Referee #3 · 26 Aug 2020

This study presents and discusses a computationally inexpensive method to build a very large data set of high impact typhoon events. The availability of this data set allows improving the assessment of the risk associated with the occurrence of typhoon landfall in the Western North Pacific area. The method is based on the application of the impact-oriented tracking algorithm named WiTRACK, adapted to the tracking of Tropical Cyclones (TCs). When applied to the THORPEX Interactive Grand Global Ensemble (TIGGE) archive, the presented method allows creating a database of non–realised typhoon event–data equivalent to more than 10,000 years of TC events. The spatial and temporal characteristics of these non–realised but physically possible events are consistent to the historical typhoon climatology in the WNP, as obtained from the JRA-55 reanalysis. Furthermore, it is shown that the created data set con-

tains up to about 100 times more very–severe and violent typhoons than the historical record and that, consequently, it allows a more reliable and less uncertain estimate of the return period–return level of TC associated extreme wind events. General Comment I think that the topic of this study is of great interest and relevance, and that it is suitable to NHESS. Besides, the paper is generally well written, the methodology clearly illustrated and the results well presented and discussed. However, there are also a few (minor) corrections and some improvements of the text that could be made to further improve the manuscript before to proceed with its publication. Therefore, my recommendation is to accept the manuscript for publication after minor revisions. Specific Remarks 1. Page 1, line 14: "... characteristics of the new event set is consistent to the..." should read "... characteristics of the new event set are consistent to the..." 2. Page 1, line 24: "... 67.1 billion RMB ..." Many readers could be helped to understand the economic significance of this figure by accompanying it with the corresponding value in US Dollars or Euros. 3. Page 2, line 45: "... (ii) the storms in the typhoon event set might not be physically consistent." Please, clarify what do you exactly mean here with "physically consistent"? 4. Page 2, line 63–64: "In this study, we show the TPEPS event set has much higher information content: more TC events and more extremely high impact TC events." Higher and more than what? 5. Page 4. Line 126: "WiTRACK identifies windstorm events of all kind, including MEPS TCs, PEPS TCs, MEPS extratropical cyclones." I suppose it identifies also PEPS extratropical cyclones. 6. Page 4, line 175: "The removal of these events ensures the TPEPS event set is independent of any pre-existing weather patterns." The goal here is to build a large set of typhoon events in order to provide a solid statistical evaluation of their characteristics, so why is it so important that the considered TPEPS events are independent of any pre-existing weather patterns? 7. Page 6, line 193, Figure 1: please add the units to the colour bar. 8. Page 6, line 197, Table 5: Why there is such a large difference in the number of simulated TC wind storms between the TIGGE models? Is this due to the different number of ensemble members of the EPSs? The large majority of the considered TPEPS are from two EPSs: the ECMWF and the

NCEP. What consequences could this fact have on the analysis results? 9. Page 6, line 202: Fig. 1d, should read Fig. 2d. 10. Page 6, line 203: Fig. 2 should read Fig. 3. 11. Page 6–7, line 212–220: I'm not sure I fully understand the explanation the authors provide for the discrepancy between the spatial distribution of the TPEPS event set and JRA–55 events as shown in Figure 2 (panels c and f). The fact that the JRA-55 event set can be considered as a subset of the TIGGE event set does not explain the difference in spatial distribution. According to this view, in fact, the JRA-55 events can be seen as randomly selected from a larger set (the TIGGE set), and thus they should also be spatially distributed as this event set. Also, why the higher level of the 98th percentile values of the JRA-55 wind should explain the lower number of typhoons in this area? 12. Page 7, line 248–249: As formulated here, this sentence seems to imply that TCs with weaker winds are also less spatially extended, which is not true. 13. Page 7, 252–255: "... impact (Befort et al., 2020). Many of the low impact TCs ..." should probably read "... impact (Befort et al., 2020), many of the low impact TCs ...". 14. Figure 8: In the text of the manuscript, there are references to panels labelled with letters (a, b, ... f), but the panels in Figure 8 are not labelled. 15. Page 9, line 317: "... based on minimisation of the root-mean-square-error (RMSE) of ...". Of what?

Please also note the supplement to this comment:
https://nhess.copernicus.org/preprints/nhess-2020-74/nhess-2020-74-RC3-supplement.pdf

---

## Author Comment (AC1) · 21 Oct 2020

**Response to Reviewer 1's comments**

We thank Reviewer 1 for the time to go through our manuscript in details. This manuscript describes a new and efficient method to produce a physical TC event set in the western North Pacific basin. In general, reviewers think after careful revision, the results of this study is of great interest and relevance, and it will be a useful contribution to the field of TC occurrence risk assessment. Here is our point-to-point response to Reviewer 1's comments.

*Response to Major Issues*

*You have explained that you only used 6-hourly surface wind speed (i.e., lines 67-69 and discussed in Section 4.2 as well), is your risk assessment model able to comprehend a full TC risk? What about TC-rainfall (you only mentioned about it in the last 2-3 sentences in Conclusions) and TC-induced storm surge, which are more significant than winds in many regions of the world. For example, the recent Hurricane Florence in USA and Mangkut in South China caused significant damaged due to surge and rainfall, respectively. Therefore, under such circumstances, what is the applicability of your proposed approach? You need to think about the generalization, replicability, and adoptability of your approach from a wider perspective and not only from the study area.*

We are thankful for the reviewer's comment to the overall impact of Typhoons and their differentiated reasoning in form of the different meteorological variables concerned. In this study, we present a method, which addresses the basic, critical issue in typhoon risk assessments – a robust methodology to determine the real frequency of a tropical cyclone (TC) occurrence with high socioeconomic impact *potential*. We are doing so by automatically identify and tracking severe, damage relevant tropical cyclones in a data set being representative of several thousand years of "observations", the TIGGE archive. Based on the large amount of data from the different multi-model multi-member ensembles to be analysed, in this study, we use thus a computationally inexpensive approach: by applying an impact-oriented tracking algorithm WiTRACK (Leckebusch et al., 2008; Kruschke, 2015; Befort et al., 2020) on multi-model ensemble forecasts to generate a large, physical consistent TC event set. This *identification* of major events is thus based on one meteorological variable only (wind speed), but is capable of identifying events of *general* loss relevance. We recently demonstrated in Befort et al. (2020) that the tracking algorithm used in this study (WiTRACK), can automatically identify the relevant (over 90%) of TCs with high overall socioeconomic impact (e.g., above 3,000 million RMB or 440 million US$ to mainland China). This implies the event set generated by our approach is in principle suitable for general TC risk assessments, as well as for an assessment of the hazards frequency-intensity distribution specifically. As we fully agree with Reviewer 1 that TC-rainfall and TC-induced storm surge are also important factors in assessing the full impact (risk) of TC to society, we would like to point out that the events identified and used in this study are including those where potential loss is caused not only by the direct wind force but also by secondary natural hazards, e.g. flooding by precipitation, or potential storm surge by the combined wind and pressure impact.

Therefore, the applicability of our proposed approach is directly visible and fundamental. While studies like e.g. Sajjad and Chan (2019) are focussing on an overall assessment of integrated risks (hazard x vulnerability/resilience; further comments to this topic, please find below) based on observed tracks from only some 60-70 years (of inhomogeneous data of different quality), our approach utilizes data (in this case multi-model synoptic forecasts) representative of ~10k years. To quantify robustly the probability of occurrence of e.g., a 100-year event from 60-70 years of observations is widely impossible (because it may potentially not have realised during this short observational period). Consequently, such an approach is not used e.g., in industry applications and is only of limited use for risk assessments as the **underlying** hazard component probability distribution is not sufficiently known.

The reviewer mentioned the topic of "*generalization, replicability, and adoptability*". We fully agree and this is exactly the reason why we developed this approach.

- By using an impact-based identification of the natural hazard relative to the datasets typical characteristics (i.e., the $98^{th}$ percentile of surface-near wind speeds), the assessment of the hazard is accounting of potential biases in the data set under investigation (e.g., AOGCM, NWP, or RCM simulations) and is linked to overall losses out of those events relative to the climatological distribution of these data sets.
- By utilizing ensemble forecasts (i.e., TIGGE's unrealised TC events), we secure the statistical robustness of our intensity-frequency distribution assessment to create an event-set being representative of the risk of occurrence of events of potentially damaging characteristic (so called tail events).

We have demonstrated the applicability and transferability of this approach in several studies for different hazards and different regions (e.g. Leckebusch et al., 2008; Nissen et al., 2014; Osinski et al., 2016; Befort et al., 2015; Befort et al., 2016; Befort et al., 2019; Walz and Leckebusch, 2019)

A more in-depth investigation of the contribution of different drivers (e.g., extreme precipitation and storm surges) of loss and damage during a (non-realised) severe TC event would be beyond the scope of this study. Nevertheless, once a representative TC event set is derived, which provides robust information of the frequency of high impact TCs, the impact of extreme precipitation and storm surges could be integrated e.g., following the approach developed and published by us in Befort et al. (2015). **We have included the above information in Sect. 5 summary for clarity.**

*Recently, Sajjad and Chan 2019 and Sajjad et al. 2020 proposed typhoon risk frameworks based on TC hazard (wind-based similar to yours), vulnerability, and disaster resilience, which provides a comprehensive information on TC risk. They found that the Pearl River delta region in Guangdong (area primarily mentioned in your case, Line) is a statistically significant hotspot of TC risk. How do you see the usefulness of your method for such frameworks? A thorough discussion regarding this is necessary*

*Similarly, most of the discussion in the manuscript revolves around TC-hazard and neglects the vulnerability and resilience within the regions where TCs are making landfalls. For instance, you say on Lines 236-238 that overall impacts of a storm is related to many factors such as size, duration, and intensity. However, the impacts are not only related to TC-*

*associated factors but vulnerability and resilience are also integral parts of overall impacts and risks associated with TCs, as discussed in Sajjad and Chan 2019 and Sajjad et al. 2020. How do you incorporate these characteristics within the TC risk discussion of yours?*

We think Reviewer 1 may have misread the purpose of our study because we do not share the same definition of the term "risk". The term "risk" in this study refers to the possibility of occurrence of an event (e.g. Vickery et al., 2000; Emanuel et al., 2006), as outlined in the title. In the context of the papers cited by Reviewer 1, this manuscript focuses on the hazard component in the framework of a classical impact modelling and assessment approach. Please note that in terms of an impact modelling approach, the components would be the hazard (and its risk of occurrence), the vulnerability (which would include measures of resilience), and the exposure of values at risk. **We have replaced the term "risk" by "hazard" in relevant places in the manuscript to make this point clearer**. To avoid confusion, hereinafter, we use the term "hazard" to represent the possibility of occurrence of a natural event; and the term "risk" to reflect a systemic integrated perspective of resultant impacts by the combination of information about the hazard, the vulnerability, and the exposure. This terminology would be more similar to the approach used in Sajjad and Chan (2019) and Sajjad et al. (2020), and will also account for the common practice in industry applications (e.g., CAT models).

*Additionally, you need to detail the current limitations of your method. For example, how well this method could perform at higher resolution assessments, which are more important for policy and decision-making in the context of DRR efforts? What are the future prospects of your study?*

Many thanks for this comment. In comparison to other methods to generate large TC event sets, our specific approach is **limited** mainly by the source of data used. The current TC event set constructed on synoptic scale forecasts archived in TIGGE, is strictly spoken representative only for the current climate state. Any longer-term climate variability (e.g. multi-decadal fluctuations like the PDO) and their impacts on any TC frequency-intensity distribution are not accounted for in this setting. Nevertheless, the presented approach would be equally applicable to data sets representing that kind of variability on longer time scales (e.g. decadal predictions or transient climate model simulations). Another limitation is obviously that we do not account for a direct assessment of the damage (loss) contribution of individual meteorological variables (e.g., precipitation leading to flooding, as mentioned above). **We added a specific section on limitations in Sect. 5 [Lines 440-446].**

It is not fully clear what Reviewer 1 refers to as "**higher resolution assessments**". The TIGGE archive provides forecast data on a spatial scale (~0.56° - 1.25°), which is not matched by any other data source of comparable length (equivalent to 10k years of observations of TCs). Further, we intentionally linked our (forecast) model based assessment to in-situ point observations from stations: the ultimate downscaling test. As we have demonstrated in section 4.3, one of the potential applications of our event set is to improve the return-period/return level calculation of the wind hazard at the local scale. Wind speed values are used in practice

to decide on payments out of e.g. parametric insurance products (Swiss Re, 2016). Consequently, reliable wind-based trigger points of typhoon parametric insurance can be determined. This will further improve the suitability and flexibility of parametric insurance for DRR applications. Ultimately, this will improve the speed of post-disaster recovery. **We added a respective paragraph to the discussion [Lines 359-363, Lines 468-476]**.

With regard to **future prospects** of this study, we discuss this in Section 5. In particular, unlike event sets generated from a stochastic approach, the TC-associated precipitation field is simulated directly by the model. This means a more complex compound TC hazard assessment, as mentioned above (limitations), can be done as well in principle. The event set that we have constructed contains all necessary information for applications in the DRR context. Once robust trigger points for the local hazard are available (including their uncertainty), the targeted application of parametric products in disaster relief application is possible. Especially, when it comes to the evaluation of the basis risk (the risk not covered by payments out of a parametric product). This study is merely the first step toward a statistically robust, full physical model based TC hazard assessment. **We added a respective paragraph to Sect. 5 [Lines 468-476]**.

**Specific Comments**

***Lines 10-13:*** *The sentence is long and it is difficult to follow. Would be better to break it into two sentences, if possible.*

We thank Reviewer 1 for pointing this out, **we have modified this sentence accordingly**.

***Line 22:*** *There is no such thing as "natural disaster" but only natural hazards. Disasters always involve human agency. Therefore, please avoid using this term and check the manuscript thoroughly for this issue. For further details, you are encouraged to see https://www.undp.org/content/undp/en/home/blog/2017/5/18/Natural-disasters-don-t-exist-but-natural-hazards-do.html*

Many thanks for pointing this out. Although we agree in principle that the disaster aspect includes a men-made perspective in its impact on human influenced structures the personal position expressed in this blog is document of a narrow understanding and perspective of nature. A natural hazard can be a disaster for the environment, even without human influences. Following good scientific practice, we prefer to use peer-reviewed literature for scientific studies and not personal comments. We used the phrase "natural disaster" as a generic term to indicate a natural event with sudden, large negative economic or environmental losses. The definition of "disaster" that we used here is similar to the definition stated in the IFRC webpage (see https://www.ifrc.org/en/what-we-do/disaster-management/about-disasters/what-is-a-disaster/). We are not trying to argue whether "natural disaster" exists or not as this is beyond the scope of this study. However, the use of this phrase is in line with literature, e.g. Cavallo and Noy (2011), Smith and Matthews (2015), Ye et al. (2016), Bakkensen et al. (2018). Nevertheless, to avoid any confusion and as we are in general agreement with the reviewer **we rephrased line 22 to: " … lead to an increase of risk to humans and for economic loss potentials from natural hazards e.g., tropical cyclones, with potentially disastrous**

**consequences." We also checked the whole manuscript and corrected for a more precise use of the terminology in the revised manuscript.**

*Line 34: you mean "livestock"?*

We thank Reviewer 1 for pointing this out and the reviewer is correct. **We have corrected this**.

*Lines 145-149: It is mentioned that VIF is used to resolve the issue of collinearity and 17 variables are selected to construct the final LRC model. How many total variables were included initially? Are the VIF values for all of these 17 variables less than the normal threshold (i.e., VIF ≤ 7.5)? It would be useful to add the VIF values of the final variables in Table 3.*

The list of variable initially used is presented in Table 2. We have modified the caption of Table 2 for clarification.
Yes, those 17 variables stated in Table 3 have VIF value of less than 5. We are not sure whether including the VIF values of the final variables would be useful for this manuscript. However, Reviewer 1 pointed out that we did not state the criteria which we used for the variable selection. **We have included the criteria (i.e. VIF < 5) for the variable selection in the manuscript. Line 176-177:** "*Variables with VIF value larger than 5 are excluded*."

*Lastly, a thorough intermediate level editing is recommended to remove "several" grammatical and language errors throughout the manuscript.*

We thank Reviewer 1's advice, and **we have edited the manuscript accordingly**.

**References**

Bakkensen, L. A., Shi, X., and Zurita, B. D.: The Impact of Disaster Data on Estimating Damage Determinants and Climate Costs, Economics of Disasters and Climate Change, 2, 49-71, 10.1007/s41885-017-0018-x, 2018.

Befort, D. J., Fischer, M., Leckebusch, G. C., Ulbrich, U., Ganske, A., Rosenhagen, G., and Heinrich, H.: Identification of storm surge events over the German Bight from atmospheric reanalysis and climate model data, Natural Hazards and Earth System Sciences, 15, 1437, 2015.

Befort, D. J., Wild, S., Kruschke, T., Ulbrich, U., and Leckebusch, G. C.: Different long-term trends of extra-tropical cyclones and windstorms in ERA-20C and NOAA-20CR reanalyses, Atmos Sci Lett, 17, 586-595, 10.1002/asl.694, 2016.

Befort, D. J., Wild, S., Knight, J. R., Lockwood, J. F., Thornton, H. E., Hermanson, L., Bett, P. E., Weisheimer, A., and Leckebusch, G. C.: Seasonal forecast skill for extratropical cyclones and windstorms, Q J Roy Meteor Soc, 145, 92-104, 10.1002/qj.3406, 2019.

Befort, D. J., Kruschke, T., and Leckebusch, G. C.: Objective identification of potentially damaging tropical cyclones over the Western North Pacific, Environmental Research Communications, 2, 031005, 10.1088/2515-7620/ab7b35, 2020.

Cavallo, E., and Noy, I.: Natural Disasters and the Economy — A Survey, International Review of Environmental and Resource Economics, 5, 63-102, 10.1561/101.00000039, 2011.

Emanuel, K., Ravela, S., Vivant, E., and Risi, C.: A statistical deterministic approach to hurricane risk assessment, B Am Meteorol Soc, 87, 299-314, 10.1175/Bams-87-3-299, 2006.

Kruschke, T.: Winter wind storms: Identification, verification of decadal predictions, and regionalization, Doktors der Naturwissenschaften, Institut f• ur Meteorologie, Freie Universit• at Berlin, 181 pp., 2015.

Leckebusch, G. C., Renggli, D., and Ulbrich, U.: Development and Application of an Objective Storm Severity Measure for the Northeast Atlantic Region, Meteorologische Zeitschrift, 17, 575-587, 10.1127/0941-2948/2008/0323, 2008.

Nissen, K. M., Ulbrich, U., Leckebusch, G. C., and Kuhnel, I.: Decadal windstorm activity in the North Atlantic-European sector and its relationship to the meridional overturning circulation in an ensemble of simulations with a coupled climate model, Climate Dynamics, 43, 1545-1555, 10.1007/s00382-013-1975-6, 2014.

Osinski, R., Lorenz, P., Kruschke, T., Voigt, M., Ulbrich, U., Leckebusch, G. C., Faust, E., Hofherr, T., and Majewski, D.: An approach to build an event set of European windstorms based on ECMWF EPS, Nat. Hazards Earth Syst. Sci., 16, 255-268, 10.5194/nhess-16-255-2016, 2016.

Sajjad, M., and Chan, J. C. L.: Risk assessment for the sustainability of coastal communities: A preliminary study, Science of The Total Environment, 671, 339-350, https://doi.org/10.1016/j.scitotenv.2019.03.326, 2019.

Sajjad, M., Chan, J. C. L., and Kanwal, S.: Integrating spatial statistics tools for coastal risk management: A case-study of typhoon risk in mainland China, Ocean & Coastal Management, 184, 105018, https://doi.org/10.1016/j.ocecoaman.2019.105018, 2020.

Smith, A. B., and Matthews, J. L.: Quantifying uncertainty and variable sensitivity within the US billion-dollar weather and climate disaster cost estimates, Nat Hazards, 77, 1829-1851, 10.1007/s11069-015-1678-x, 2015.

Swiss Re: Natural catastrophes and man-made disasters in 2015: Asia suffers substantial losses. https://reliefweb.int/sites/reliefweb.int/files/resources/sigma1_2016_en.pdf, 2016.

Vickery, P. J., Skerlj, P. F., and Twisdale, L. A.: Simulation of Hurricane Risk in the U.S. Using Empirical Track Model, Journal of Structural Engineering, 126, 1222-1237, 10.1061/(ASCE)0733-9445(2000)126:10(1222), 2000.

Walz, M. A., and Leckebusch, G. C.: Loss potentials based on an ensemble forecast: How likely are winter windstorm losses similar to 1990?, Atmos Sci Lett, 20, e891, 10.1002/asl.891, 2019.

Ye, T., Wang, Y., Wu, B., Shi, P., Wang, M., and Hu, X.: Government Investment in Disaster Risk Reduction Based on a Probabilistic Risk Model: A Case Study of Typhoon Disasters in Shenzhen, China, International Journal of Disaster Risk Science, 7, 123-137, 10.1007/s13753-016-0092-7, 2016.

---

## Author Comment (AC2) · 21 Oct 2020

**Response to Reviewer 2's comments**

We thank Reviewer 2 for the time to go through our manuscript in details. This manuscript describes a new and efficient method to produce a physical TC event set in the western North Pacific basin. In general, reviewers think after careful revision, the results of this study is of great interest and relevance, and it will be a useful contribution to the field of TC risk assessment. Here is our point-to-point response to Reviewer 2's comments.

*General comments:*
*This manuscript describes a new and efficient method to produce TC event set in the Pacific basin. The TC events are detected from an ensemble data archive TIGGE, using an objective impact-oriented windstorm identification algorithm WiTRACK. This dataset contributes to existing synthetic datasets (mostly statistical basin-wide methods) as in this dataset the TCs are detected in GCMs, so that the complex physical processes of TCs are captured and hence TCs are physically realistic. More extreme TCs are found in the dataset and these data will help overcome the shortage in observational record. Overall, I think the results will be a nice contribution to the field of TC risk assessment. However, I have some basic questions about the methodology, the utility of certain results and I recommend major revisions prior to publication. Also, I would recommend careful editing of the manuscript. There are many terminologies in the manuscript and please make sure it is easy for readers to follow.*

We thank the reviewer advice and we have carefully reviewed and edited the manuscript.

*Major comments:*
*1) L191: The detection rates of historical TCs are reported here. However, will the detection algorithm produce more TCs? What fraction of TCs that the detection algorithm produce is real historical TCs?*

We thank the reviewer for pointing this out and highlighting a section for which we can improve clarity. Strictly spoken, the detection algorithm we apply in this study (developed for TC detection in the West-Pacific in Befort et al., 2020) does not produce TCs, but enables us to detect them automatically in the large data set. As the data set we use is the output of operational NWP's forecast models, in a very narrow sense , none of the TCs that we detected is a one-to-one equivalent to a real historical TCs. However, events which satisfy the criteria in the MPES TC identifier (MTI, Section 3.2.3) (i.e. MEPS TC events) can be considered as events which are similar to the historical event. The percentage of TCs which are in this sense similar to real events that occurred in the TIGGE is ~60%. Thus, about 40% are pure ensemble predicted events that did not realise in the observed nature or do not have a very similar twin at the same time at the same location.

**We have modified the text in L191 and the caption of Table 4 for more accurate description. Lines 225-228**, "*A historical TC is said to be detected in a forecast model if there exists a TC counterpart in the forecast model, which is similar to the historical TC as identified by the MTI (Section 3.2.3). The detection rates of historical TCs which are detected in different*

*forecast outputs, i.e. CMA, ECMWF, JMA, and NCEP, are 91.2%, 94.7%, 89.4%, and 90.7%, respectively, ...*"

*2) The authors mention that one benefit of this dataset is "The TPEPS event set includes events which are unlikely but physically possible. This provides an important and unique advantage for typhoon risk assessment." Combined with Fig. 2, TC tracks in the detected dataset is very different with observations, and TPEPS tracks appear in locations with no historical tracks. If there is no historical track in some regions, are they supposed to be no storms or there can be storms but no storm has appeared in historical records due to the low probability? This needs to be explained.*

We thank Reviewer 2 for pointing out this important issue. If there is no historical track in some regions, this does not mean storms cannot occur in those regions. The fact that we have not seen a TC during the time period of known observational records in those regions could be due to the observation period is too short and the sample size is not large enough to fully represent the distribution of the underlying basic population (i.e. all possible TCs in the given climate state). For example, if we follow the necessary but insufficient conditions of TC formation which are identified by Gray (1977) from historical observations, TC formation occurs away from the equator (> 5 deg). However, Tropical Storm Vamei (2001) formed close to the equator (~1.4 deg N). This shows storm can appear in the historically "storm-free" region.

Furthermore from the statistical perspective, we can view the JRA-55 event set as a subset which is randomly selected from the TPEPS event set. This means if we randomly sample the TPEPS event set, we can obtain a subset highly similar to the JRA-55 event set. For demonstration, we have conducted bootstrap resampling on the TPEPS event set to obtain 10,000 sets of subsample. Each set of subsamples has 668 events to mimic the number of events in the JRA-55 event set. For each set of subsamples, the track density is calculated, and used to calculate uncentred pattern correlation between the resampling set of subsamples and the JRA-55 event set. In order to focus on relevant entries, for a particular grid box, if the values of track density for a resampling set and the JRA-55 event set are both less than one, such grid box is not used in the pattern correlation calculation. The mean, standard deviation, minimum, and maximum of the uncentred pattern correlation of the 10,000 set of subsamples are 0.9380, 0.0107, 0.8961, and 0.9697, respectively. This suggests the spatial pattern of the JRA-55 event set is highly similar to a small random subset of the TPEPS event set. Consequently, the JRA-55 event set can be seen as a subset randomly selected from the TPEPS event set. On the other hand, it is not be possible to deduce the basic population (e.g. the TPEPS event set) from a small sample set (e.g. the JRA-55 event set). Although the spatial distribution of the small set sample is similar to the subsamples of the basic population and thus usable as one possible realisation of the basic population, the small sample set does not contain all of the information of the underlying population. Furthermore, the statistical estimate of extremes would also be different for the small sample set (e.g. JRA-55 event set) and the basic population (e.g. TPEPS event set).
**We have included the above explanation in the revised manuscript (Lines 259-280).**

*3) The sensitivity and performance of four ensemble data archive are not well described. For example, in some dataset, the storms are much weaker than historical storms. And some models*

*have biases in simulating extratropical cyclone transition. More explanations and descriptions of the data archive needs to be added. Also, how these biases would have an impact on the detection algorithm?*

The four data sets selected from the TIGGE archive are the state-of-the-art NWP models as used by four leading synoptic weather forecast centres worldwide. Although a full assessment of their respective models' skill and potential biases is not in the scope of this study, **we added a section with information on the general performance of these four selected NWP models (Lines 106-116)**. For the dedicated purpose of this study, the reviewer is fully correct and we need to check for biases in the underlying climatological features as provided by a time- and ensemble-aggregated view of the data set (a task normally not necessarily done in forecast model evaluation departments for all levels of severe and rare extremes). This evaluation for extreme TC occurrence is what we did in section 4.1, showing respective results in Fig.1-7. **We included a paragraph to clarify which part of the study is model validation and which part is event set building (Lines 223-225)**.

TIGGE data's main difference to the operationally used NWP output is that TIGGE did archive a lower resolution. Nevertheless, all underlying processes and feedbacks are captured in the originally resolution of the NWP products and are thus fully included. Thus, we would expect the best possible representation of dynamical processes in those forecast simulations than compared to lower resolution AOGCM simulations, e.g. for transient climate experiments. Beyond this, model resolution is known to be a limiting fact of simulating TC intensity (Bengtsson et al., 2007). One of the advantages of using WiTRACK is that it does not use raw wind speeds, instead, it uses the 98$^{th}$ percentile relative exceedance for tracking. This means that even if the simulation wind speed of TC is systematically weaker than in historical observations, the 98$^{th}$ percentile climatological wind should also be lower than the actual 98$^{th}$ percentile climatological wind speed, a TC will still be tracked as long as there exists a 98$^{th}$ percentile exceedance wind cluster. It can be shown that, within the study area, the 98$^{th}$ percentile relative exceedance of the 4 models, which we used to construct the TIGGE event set, have similar behaviour (i.e. similar to Figure 2 of Osinski et al. (2016)). Befort et al. (2020) showed the applicability of such an approach to relate information from observations (i.e. IBTrACS data) to automatically detected TCs from a much coarser resolution reanalysis product (JRA-55). Consequently, a bias due to resolution does not have significant impact on WiTRACK as the tracking algorithm serves as a bias correction in this sense (detailed discussion on the impact of weaker wind speed in model outputs on WiTRACK can be found in Osinski et al. (2016)). **We included a paragraph to discuss this in more detail (Lines 126-128; 141-152)**.

*4) The authors have compared the TIGGE PEPS TCs with JRA-55 in terms of track density, landfall frequency, etc. How about other characteristics? For example, landfall intensity along coastline?*

The distribution of landfall intensity (wind speed in m/s) for TC, which made landfall with at least typhoon strength, are very similar for the JRA-55 event set and TPEPS event set. The table below shows some of the statistics of these two distributions. The two-sample Kolmogorov-Smirnov test show these two distributions belong to the same distribution significant at the 0.05 significance level.

| | JRA-55 | TPEPS |
|---|---|---|
| Mean | 23.5899 | 23.4044 |
| Standard deviation | 3.44527 | 3.84537 |
| Median | 22.58 | 22.2 |
| Number of events | 184 | 23343 |

**We have included the above discussion into revised manuscript [Lines 344-347].**

*5) Fig. 7 shows the difference between TIGGE PEPS event set and observation. In the text, you have mentioned possible reasons for these differences. Is there possible way to reduce these differences, for example in the detection algorithm, to also remove low-impact storms? Also, you mentioned the ESSI, is there a way to quantify this index?*

Figure 7 shows some of the differences between the TPEPS event set and the JRA-55 event set. These differences are mainly due to the finite simulation time in forecast models. Some of these differences could be reduced based on additional assumptions that would depend on the specific application of the users. A more detailed analysis of the performance with respect to a data set not affected by a finite simulation time is a reanalysis product (e.g. JRA-55). We showed in Befort et al. (2020) in JRA-55 that our tracking already focusses on the most severe part of the TC severity distribution and thus does show some expected differences to e.g. IBTrACS data.

We apologise we did not include the text associated with the SSI (and ESSI). Leckebusch et al. (2008) introduced this objective severity measure for gridded datasets of extreme storms in the North-Atlantic and the method was applied for TCs in the North-West Pacific in Befort et al. (2020). **The relevant text is included in revised manuscript [Lines 215-219]**.

***Minor comments:***
*L41-42: more recent papers should be added. Such as the following two recent models:*
*- Lee, C.-Y., M. K. Tippett, A. H. Sobel, and S. J. Camargo, 2018: An environmentally forced tropical cyclone hazard model. Journal of Advances in Modeling Earth Systems, 10 (1), 223–241.*
*- Jing, R., and N. Lin, 2020: An environment-dependent probabilistic tropical cyclone model. Journal of Advances in Modeling Earth Systems, 12 (3), e2019MS001 975.*

We thank the reviewer's suggestions. **We have included these references in the revised manuscript.**

*L45: I didn't understand the sentence 'the typhoon event set might not be physically consistent'. What is 'physically' consistent?*

It means event sets created by stochastic perturbations will create TC events that (with respect to their inner dynamical structure) are not necessarily physically consistent anymore. As just surface footprints are stochastically modelled from existing tracks, there is no check whether those events (in the stochastically modelled from) are physically possible and how they could

be realised in a fully dynamical consistent view, thus fulfilling all known physical relations and derived constraints by the means of physical laws. Consequently, the amount of unrealistic physical properties due to the oversimplified stochastic simulation is unknown and laws of physical interactions are potentially ignored. **We have modified the sentence in the revised manuscript to clarify this point [see lines 46-52].**

*L79: "The domain of this study covers the Western North Pacific (WNP), east and south-east Asia spanning from 85 E to 195E and 15 S to 75 N." Why data around equator is also used? There is no TCs forming around equator.*

We thank Reviewer 2 for pointing this out and we apologise for the confusion. The domain stated in the manuscript is part of the parameter set up for WiTRACK. However, the true domain which is used for tracking is 90-180° E, and 0-70° N and **we have made this correction in the revised manuscript.**

We included regions close to the equator although TCs rarely form around the equator, it is still possible for TCs to form close to the equator, for example Tropical storm Vamei (2001). Furthermore, while the core pressure centre of the TCs might be away from the equator, the damaging wind field, as identified by the 98$^{th}$ percentile relative exceedance, could be quite large, impacting potentially regions close to the equator.

*L102-104: Is there a reason why an old version of IBTrACS is used?*

IBTrACS v03r10 was the most up-to-date official version of IBTrACS when this study was first started. Furthermore, for our study period (with 6-hourly observations), the data in v04 and v03r10 are the same.

*L152: "the accuracy of the LRC is about 90%" What is the fraction of TC (or positive samples?) Does there exist issue of imbalanced data?*

No, the validation set is not imbalanced. In the validation set, 49 out of 96 tracks are TCs (~51% of the validation set). **We have included a more detailed description in the revised manuscript. Lines 180-181** "*Validation using JRA-55 event set (2015-2017), which has 49 TC events and 47 non-TC events…*"

*L197: "Percentage of total TC windstorms as PEPS TCs can be treated as a proxy to quantify the forecast skill of the model." In Table 5, NCEP is almost twice of that in JMA, what does this percentage mean?*

This indicates the NCEP model generates more "wrong" forecasts than JMA yet these wrong forecasts are physically possible. **We included a clarifying sentence to a respective possible interpretation at lines 235-238**: "*For example, NCEP has 47.1% of TC windstorms as PEPS TCs whereas JMA has 26.5%. This indicates the NCEP model generates more "wrong" forecast than JMA however these wrong forecasts are physically possible. Yet, examining the*

*forecast skill of models is not the focus of this study and the rest of the discussion focuses on the TPEPS TC event set.*"

*L203: do you mean Fig 3? Also, more explanations should be added in the text. I can't understand this figure.*

We thank Reviewer 2 for identifying this error. The reviewer is correct that we are referring to Fig 3. Fig. 3 shows the feature scaled times series of number of TCs which are first identified in each day from May to December. The core message of Fig 3 is that the temporal variability of the TPEPS event set and the JRA-55 event set are largely similar (except for the earlier years). **We have modified the text in the revised manuscript. Lines 239-240** "*Figures 2 and 3 show the spatial pattern and temporal variability of the number of TC which are first detected for each day, …*"

*L203: In Fig. 2, all TPEPS are much more similar with each other, comparing with JRA-55. How to explain this?*

The major difference between the track density of TPEPS and JRA-55 is that there is an eastward bias in the TPEPS. There are several reasons that could contribute to this. The eastward bias in the track density appears to be a common feature in many GCMs (e.g. Camargo et al., 2005; Bell et al., 2013; Roberts et al., 2020), this has also been observed in seasonal forecast output (Camp et al., 2015). Finite simulation time has also contributed to this bias as TC that forms in the region east of 150 °E would not have the time to move into the western part of WNP before the end of simulation time. Differences in the amount of tracks could also contribute to the differences as more diverse tracks would be captured. **We have added a respective explanatory comment at lines 252-258**.

*Fig4: The tracks in black are very easily messed up with the map. Probably change the color of coastline.*

We thank Reviewer 2 for pointing this out. **We have changed the colour of the plot**.

*Fig5: The y-axis is not clear to me, please add more explanation.*

Fig 5 shows the climatological seasonal cycle of TC activity for the TPEPS TC event set and the JRA-55 event set. The daily number distribution, $p_i$, is calculated as follows:

$$p_i = \frac{n_i}{\sum_i n_i} \times 100\%$$

where $n_i$ is the number of TC first detected on day $i$ for the individual event set. As such, it is the probability of TC being first detected at a given day**. We have added more explanation in the caption of Figure 5.**

*Fig8: The colored dots for single center are too light to see. If this figure is to show distribution, I would recommend not using same color bar for single model and for TIGGE total.*

**We have changed the colour scale of this figure in the revised manuscript**.

*Fig9: It's hard to see the distributions are in good agreement, probably can change to annual frequency instead of total number of landfall events. Also, the correlation coefficients could be used to show the landfall frequency in all TIGGE dataset is positively correlated with JRA-55.*

We thank Reviewer 2 for these suggestions. **We have included pattern correlation between the spatial distribution for JRA-55 and TPEPS event sets in the revised manuscript. Line 338** "*…with uncentred pattern correlation of 0.8345.*"

*Fig12: I can see your points in showing the grey dashed lines. But the lower bound curves cannot show the trend properly. I would recommend add 75% or 80% confidence interval to show that the trends are same, but TIGGE PEPS event set has much narrower bounds.*

We are not certain what Reviewer 2 refers to as the trend of the lower bound curves. There are two separate factors that determine the "shape" of the curve of the lower and upper bound of uncertainty. First, the return level-return period estimate has asymptotic behaviour. This means the return level estimate approaches to a certain value as the return period increases. Second, the uncertainty of the estimation increases with increasing return period. Combining these two factors we can see that the so-called "trend" in the lower bound grey curve does not exist. To show the 95% confidence interval reflects a typical setting for assessing statistical estimates uncertainty for GPD fitted return-level plots and the authors would prefer to stay with this representation.

**References**

Bell, R., Strachan, J., Vidale, P. L., Hodges, K., and Roberts, M.: Response of Tropical Cyclones to Idealized Climate Change Experiments in a Global High-Resolution Coupled General Circulation Model, J Climate, 26, 7966-7980, 10.1175/JCLI-D-12-00749.1, 2013.

Bengtsson, L., Hodges, K. I., and Esch, M.: Tropical cyclones in a T159 resolution global climate model: Comparison with observations and re-analyses, Tellus A, 59, 396-416, 2007.

Camargo, S. J., Barnston, A. G., and Zebiak, S. E.: A statistical assessment of tropical cyclone activity in atmospheric general circulation models, Tellus A, 57, 589-604, 10.1111/j.1600-0870.2005.00117.x, 2005.

Camp, J., Roberts, M., MacLachlan, C., Wallace, E., Hermanson, L., Brookshaw, A., Arribas, A., and Scaife, A. A.: Seasonal forecasting of tropical storms using the Met Office GloSea5 seasonal forecast system, Q J Roy Meteor Soc, 141, 2206-2219, 10.1002/qj.2516, 2015.

Gray, W. M.: Tropical Cyclone Genesis in the Western North Pacific, J Meteorol Soc Jpn, 55, 465-482, 1977.

Osinski, R., Lorenz, P., Kruschke, T., Voigt, M., Ulbrich, U., Leckebusch, G. C., Faust, E., Hofherr, T., and Majewski, D.: An approach to build an event set of European windstorms based on ECMWF EPS, Nat. Hazards Earth Syst. Sci., 16, 255-268, 10.5194/nhess-16-255-2016, 2016.

Roberts, M. J., Camp, J., Seddon, J., Vidale, P. L., Hodges, K., Vanniere, B., Mecking, J., Haarsma, R., Bellucci, A., Scoccimarro, E., Caron, L.-P., Chauvin, F., Terray, L., Valcke, S., Moine, M.-P., Putrasahan, D., Roberts, C., Senan, R., Zarzycki, C., and Ullrich, P.: Impact of Model Resolution on Tropical Cyclone Simulation Using the HighResMIP–PRIMAVERA Multimodel Ensemble, J Climate, 33, 2557-2583, 10.1175/JCLI-D-19-0639.1, 2020.

---

## Author Comment (AC3) · 21 Oct 2020

**Response to Reviewer 3's comments**

We thank Reviewer 3 for the time to go through our manuscript in details. This manuscript describes a new and efficient method to produce a physical TC event set in the western North Pacific basin. In general, reviewers think after careful revision, the results of this study is of great interest and relevance, and it will be a nice contribution to the field of TC risk assessment. Here is our point-to-point response to Reviewer 3's comments.

*General Comment*
*I think that the topic of this study is of great interest and relevance, and that it is suitable to NHESS. Besides, the paper is generally well written, the methodology clearly illustrated and the results well presented and discussed. However, there are also a few (minor) corrections and some improvements of the text that could be made to further improve the manuscript before to proceed with its publication.*

*Therefore, my recommendation is to accept the manuscript for publication after **minor revisions**.*

*Specific Remarks*
*1. Page 1, line 14: "… characteristics of the new event set is consistent to the…" should read "… characteristics of the new event set are consistent to the…"*

We thank Reviewer 3 for pointing this out, **we have corrected this in the revised manuscript**.

*2. Page 1, line 24: "… 67.1 billion RMB …" Many readers could be helped to understand the economic significance of this figure by accompanying it with the corresponding value in US Dollars or Euros.*

We thank Reviewer 3's suggestion, **we have added the corresponding value in Euros in the revised manuscript**.

*3. Page 2, line 45: "… (ii) the storms in the typhoon event set might not be physically consistent." Please, clarify what do you exactly mean here with "physically consistent"?*

It means event sets created by stochastic perturbations will create TC events that (with respect to their inner dynamical structure) are not necessarily physically consistent anymore. As just surface footprints are stochastically modelled from existing tracks, there is no check whether those events (in the stochastically modelled from) are physically possible and how they could be realised in a fully dynamical consistent view, thus fulfilling all known physical relations and derived constraints by the means of physical laws. Consequently, the amount of unrealistic physical properties due to the oversimplified stochastic simulation is unknown and laws of physical interactions are potentially ignored. **We have modified the sentence in the revised manuscript to clarify this point [see lines 46-52]**.

4. *Page 2, line 63–64: "In this study, we show the TPEPS event set has much higher information content: more TC events and more extremely high impact TC events." Higher and more than what?*

We thank Reviewer 3 for pointing out this point. **This sentence should read "*In this study, we show the TPEPS event set has much higher information content: more TC events and more extremely high impact TC events than historical or reanalysis-based TC event set.*" (Lines 70-71)**

5. *Page 4. Line 126: "WiTRACK identifies windstorm events of all kind, including MEPS TCs, PEPS TCs, MEPS extratropical cyclones." I suppose it identifies also PEPS extratropical cyclones.*

Yes, it does. **We have added PEPS extratropical cyclones to the list for clarification**.

6. *Page 4, line 175: "The removal of these events ensures the TPEPS event set is independent of any pre-existing weather patterns." The goal here is to build a large set of typhoon events in order to provide a solid statistical evaluation of their characteristics, so why is it so important that the considered TPEPS events are independent of any pre-existing weather patterns?*

To use this as an extension of event numbers and thus as an alternative reality, the inclusion of real existing events will incorporate some bias towards observed events as all of them will create a multiple realisation in the ensemble members started at the time such a real event occurred. By not considering those ensemble members, which are closely related to observed events, will secure that indeed new events are used to build the pure EPS event set. It has to be noted though that the inclusion of those events should not change the overall track distribution, or in other words, the track distribution from pure EPS and real EPS events is fairly similar.

7. *Page 6, line 193, Figure 1: please add the units to the colour bar.*

**We have added the unit to colour bar**.

8. *Page 6, line 197, Table 5: Why there is such a large difference in the number of simulated TC wind storms between the TIGGE models? Is this due to the different number of ensemble members of the EPSs? The large majority of the considered TPEPS are from two EPSs: the ECMWF and the NCEP. What consequences could this fact have on the analysis results?*

The main reasons for differences in the number of detected TC windstorms between TIGGE models are they have (1) different numbers of ensemble members of the EPSs, (2) different number of runs per day, and (3) different maximum forecast lead time (c.f. Table 1). Given the spatial and temporal distributions of the individual PEPS event sets are similar to each other, the analysis on the overall TPEPS event set is reliable.

*9. Page 6, line 202: Fig. 1d, should read Fig. 2d.*

We thank Reviewer 3 for pointing this out, **we have corrected this in the revised manuscript**.

*10. Page 6, line 203: Fig. 2 should read Fig. 3.*

We thank Reviewer 3 for pointing this out, **we have corrected this in the revised manuscript**.

*11. Page 6–7, line 212–220: I'm not sure I fully understand the explanation the authors provide for the discrepancy between the spatial distribution of the TPEPS event set and JRA–55 events as shown in Figure 2 (panels c and f). The fact that the JRA-55 event set can be considered as a subset of the TIGGE event set does not explain the difference in spatial distribution. According to this view, in fact, the JRA-55 events can be seen as randomly selected from a larger set (the TIGGE set), and thus they should also be spatially distributed as this event set. Also, why the higher level of the 98th percentile values of the JRA-55 wind should explain the lower number of typhoons in this area?*

We agree with Reviewer 3 that the JRA-55 event set can be seen as a subset randomly selected from a larger set (i.e. the TIGGE event set). This means if we randomly sample the TPEPS event set, we can obtain a subset highly similar to the JRA-55 event set. For demonstration, we have conducted bootstrap resampling on the TPEPS event set to obtain 10,000 sets of subsample. Each set of subsamples has 668 events to mimic the number of events in the JRA-55 event set. For each set of subsamples, the track density is calculated, and used to calculate uncentred pattern correlation between the resampling set of subsamples and the JRA-55 event set. In order to focus on relevant entries, for a particular grid box, if the values of track density for a resampling set and the JRA-55 event set are both less than one, such grid box is not used in the pattern correlation calculation. The mean, standard deviation, minimum, and maximum of the uncentred pattern correlation of the 10,000 set of subsamples are 0.9380, 0.0107, 0.8961, and 0.9697, respectively. This suggests the spatial pattern of the JRA-55 event set is highly similar to a small random subset of the TPEPS event set. Consequently, the JRA-55 event set can be seen as a subset randomly selected from the TPEPS event set. On the other hand, it is **not** be possible to deduce the basic population (e.g. the TPEPS event set) from a small sample set (e.g. the JRA-55 event set). Although the spatial distribution of the small set sample is similar to the subsamples of the basic population and thus usable as one possible realisation of the basic population, the small sample set does not contain all of the information of the

underlying population. Furthermore, the statistical estimate of extremes would also be different for the small sample set (e.g. JRA-55 event set) and the basic population (e.g. TPEPS event set). **We have included the above explanation in the revised manuscript (Lines 259-280).**

Upon further investigation, we found that the 98$^{th}$ percentile is not the reason that leads to the differences in spatial distribution. The major difference between the track density of TPEPS and JRA-55 is that there is an eastward bias in the TPEPS. There are several reasons that could contribute to this. The eastward bias in the track density appears to be a common feature in many GCMs (e.g. Camargo et al., 2005; Bell et al., 2013; Roberts et al., 2020), this has also been observed in seasonal forecast output (Camp et al., 2015). Finite simulation time has also contributed to this bias as TC that forms in the region east of 150 °E would not have the time to move into the western part of WNP before the end of simulation time. Differences in the amount of tracks could also contribute to the differences as more diverse tracks would be captured. **We have added a respective explanatory comment at lines 252-258.**

*12. Page 7, line 248–249: As formulated here, this sentence seems to imply that TCs with weaker winds are also less spatially extended, which is not true.*

The impact area of a TC in this study refers to the total area which has experienced TC-associated extreme wind (i.e. larger than local climatological 98$^{th}$ percentile wind speed). Given the fact that the wind speed of TC wind field decays radially outward, TC with weaker winds would have a smaller impact area because the outer wind speed would be below the 98$^{th}$ local climatological wind percentile value. **We have added more descriptions about impact area in the revised manuscript to clarify this point [Lines 305-306].**

*13. Page 7, 252–255: "… impact (Befort et al., 2020). Many of the low impact TCs …" should probably read "… impact (Befort et al., 2020), many of the low impact TCs …".*

We thank Reviewer 3 for pointing this out, **we have corrected this in the revised manuscript**.

*14. Figure 8: In the text of the manuscript, there are references to panels labelled with letters (a, b, … f), but the panels in Figure 8 are not labelled.*

We thank Reviewer 3 for spotting this error. **We have corrected this in the revised manuscript**.

*15. Page 9, line 317: "… based on minimisation of the root-mean-square-error (RMSE) of …". Of what?*

We thank Reviewer 3 for pointing this out, **we have corrected this in the revised manuscript**. This is the root-mean-square-error of the quantile mapping output.

**References**

Bell, R., Strachan, J., Vidale, P. L., Hodges, K., and Roberts, M.: Response of Tropical Cyclones to Idealized Climate Change Experiments in a Global High-Resolution Coupled General Circulation Model, J Climate, 26, 7966-7980, 10.1175/JCLI-D-12-00749.1, 2013.

Camargo, S. J., Barnston, A. G., and Zebiak, S. E.: A statistical assessment of tropical cyclone activity in atmospheric general circulation models, Tellus A, 57, 589-604, 10.1111/j.1600-0870.2005.00117.x, 2005.

Camp, J., Roberts, M., MacLachlan, C., Wallace, E., Hermanson, L., Brookshaw, A., Arribas, A., and Scaife, A. A.: Seasonal forecasting of tropical storms using the Met Office GloSea5 seasonal forecast system, Q J Roy Meteor Soc, 141, 2206-2219, 10.1002/qj.2516, 2015.

Roberts, M. J., Camp, J., Seddon, J., Vidale, P. L., Hodges, K., Vanniere, B., Mecking, J., Haarsma, R., Bellucci, A., Scoccimarro, E., Caron, L.-P., Chauvin, F., Terray, L., Valcke, S., Moine, M.-P., Putrasahan, D., Roberts, C., Senan, R., Zarzycki, C., and Ullrich, P.: Impact of Model Resolution on Tropical Cyclone Simulation Using the HighResMIP–PRIMAVERA Multimodel Ensemble, J Climate, 33, 2557-2583, 10.1175/JCLI-D-19-0639.1, 2020.